# Altering metabolism programs cell identity via NAD⁺-dependent deacetylation

Robert A Bone[1], Molly P Lowndes [1,2], Silvia Raineri [1], Alba R Riveiro [1], Sarah L Lundregan[1], Morten Dall [3], Karolina Sulek[3], Jose A H Romero [1], Luna Malzard[4], Sandra Koigi[1], Indra J Heckenbach[4], Victor Solis-Mezarino[5], Moritz Völker-Albert[5], Catherine G Vasilopoulou[6], Florian Meier [6], Ala Trusina [4], Matthias Mann [2,6], Michael L Nielsen [2], Jonas T Treebak [3] & Joshua M Brickman [1]✉

## Abstract

**Cells change their metabolic profiles in response to underlying gene regulatory networks, but how can alterations in metabolism encode specific transcriptional instructions? Here, we show that forcing a metabolic change in embryonic stem cells (ESCs) promotes a developmental identity that better approximates the inner cell mass (ICM) of the early mammalian blastocyst in cultures. This shift in cellular identity depends on the inhibition of glycolysis and stimulation of oxidative phosphorylation (OXPHOS) triggered by the replacement of D-glucose by D-galactose in ESC media. Enhanced OXPHOS in turn activates NAD + -dependent deacetylases of the Sirtuin family, resulting in the deacetylation of histones and key transcription factors to focus enhancer activity while reducing transcriptional noise, which results in a robustly enhanced ESC phenotype. This exploitation of a NAD + /NADH coenzyme coupled to OXPHOS as a means of programming lineage-specific transcription suggests new paradigms for how cells respond to alterations in their environment, and implies cellular rejuvenation exploits enzymatic activities for simultaneous activation of a discrete enhancer set alongside silencing genome-wide transcriptional noise.**

**Keywords** Metabolism; Sirtuins; Enhancers; Pluripotency; Aging
**Subject Categories** Development; Metabolism; Stem Cells & Regenerative Medicine

## Introduction

Cells can generate energy via two principal processes: glycolysis and oxidative phosphorylation (OXPHOS). A cell's mode of energy production is frequently altered in differentiation, but it is unclear the extent to which global shifts in metabolism can directly induce cellular phenotypes. Alterations in metabolism have been shown to influence global patterns of chromatin regulation (Dai et al, 2020), but how can the simple selection of an energy source be used to influence highly specific programs of gene expression or different cell identities? The early mammalian embryo and cell lines derived from it are characterized by distinct metabolic states that have long been assumed to be a byproduct of differentiation, and together they represent excellent models to address how metabolism dictates phenotype, rather than a phenotype driving a cell's choice of energy source.

During pre-implantation development, cells are concerned with the generation of both the embryonic and extra-embryonic lineages. In mice, the segregation of embryonic from extra-embryonic fate occurs as a result of two successive cell fate choices, the first leading to segregation of the inner cell mass (ICM) and the extra-embryonic trophectoderm (TE), and the second in which the ICM undergoes further specification into the Epiblast (Epi) and the extra-embryonic Primitive Endoderm (PrE) (Riveiro and Brickman, 2020). Epi cells retain their capacity to generate somatic, but not extra-embryonic lineages, and are therefore termed pluripotent. Previous studies have shown that the nutritional history and metabolism of stem cells are influential in directing cell fate in a variety of species (Ito et al, 2012; Chattwood et al, 2013; Moussaieff et al, 2015; Ryall et al, 2015; Sperber et al, 2015; Cornacchia et al, 2019; Khoa et al, 2020; Folmes et al, 2011). Morula-stage embryos are also known to change their metabolism as they develop into a blastocyst, relying more upon glucose metabolism and increasing levels of glycolysis (Leese, 2012; Chi et al, 2020). With the specification of the ICM at the morula stage, the TE relies on high levels of both pyruvate (glycolytic metabolism) and OXPHOS, whereas the ICM is principally dependent on OXPHOS.

Embryonic stem cells (ESCs) are immortal cell lines derived from the peri-implantation blastocyst and are considered pluripotent as they can contribute to all lineages of the embryo proper. They can be cultured in standard serum-containing media with the

[1]Novo Nordisk Foundation Center for Stem Cell Medicine (reNEW), Department of Biomedical Sciences, University of Copenhagen, Copenhagen, Denmark. [2]Novo Nordisk Foundation Center for Protein Research, Department of Cellular and Molecular Medicine, University of Copenhagen, Copenhagen, Denmark. [3]Novo Nordisk Foundation Center for Basic Metabolic Research, Copenhagen, Denmark. [4]Niels Bohr Institute, University of Copenhagen, Copenhagen, Denmark. [5]MoleQlar Analytics GmbH, Rosenheimer Straße 141h, 81671 Munich, Germany. [6]Department of Proteomics and Signal Transduction, Max Planck Institute of Biochemistry, Martinsried, Germany. ✉E-mail: joshua.brickman@sund.ku.dk

cytokine LIF (Serum/LIF) or in various defined conditions, including one that exploits inhibitors of two prominent differentiation-promoting signals alongside LIF (2i/LIF) (Ying et al, 2008; Morgani et al, 2017; Riveiro and Brickman, 2020). Using a fluorescent transcriptional reporter for the early endoderm gene *Hhex*, we previously identified a subpopulation of ESCs in both 2i/LIF culture and media supplemented with knockout serum replacement (KOSR) and LIF (KOSR/LIF), that co-expressed *Hhex* with Epi markers, such as NANOG, suggesting this population has earlier unsegregated ICM-like qualities (Canham et al, 2010; Morgani et al, 2013; Martin Gonzalez et al, 2016). Consistent with this notion, ESCs that co-express Epi and PrE markers can also differentiate into both the embryonic and extra-lineages of the blastocyst, and therefore exhibit greater than pluripotent, or totipotent properties (Morgani et al, 2013; Lo Nigro et al, 2017; Redó-Riveiro et al, 2024). Although this ICM-like cell type can arise in a variety of conditions, it correlates with the enrichment of modulators of lipid metabolism (Martin Gonzalez et al, 2016).

Here, we explore the role of metabolism in regulating cellular phenotypes. We exploit altered sugar metabolism to force ESCs to upregulate OXPHOS at the expense of glycolysis, leading to increased ICM-like transcriptional identity and the generation of cultures of enhanced metabolic ESCs (EMESCs). Moreover, this state appears to be about more than ICM identity, rather an exceptionally clean transcriptional signature featuring the amplification of lineage-appropriate enhancer signatures while reducing epigenetic noise, leading to diminished phenotypic variation and improved functionality. In these cultures, the OXPHOS cofactor, $NAD^+$, triggers this activation of a lineage-specific transcriptional program via Sirtuin deacetylase activity. This simultaneous silencing of extraneous enhancers, alongside the tuning of activity at those that are lineage-relevant, produces an enhanced transcriptional signal-to-noise ratio that could be the basis for the role of these factors in aging.

# Results

## Altering the balance between OXPHOS and glycolysis reprograms ESCs to an early embryonic state

To confirm that ESC cultures supporting ICM-like cell types display a unique metabolic profile, we performed liquid chromatography-tandem mass spectrometry (LC-MS/MS). We found 468 differentially expressed proteins between all three conditions (Appendix Fig. S1A,B; Dataset EV1). In the cluster of proteins that were upregulated in cultures supporting ICM-like states (2i/LIF, KOSR/LIF), proteins involved in lipid metabolism were enriched (Appendix Fig. S1C,D). We next assessed the oxygen consumption rate (OCR) and extracellular acidification rate (ECAR), which are measures of OXPHOS and glycolysis, respectively, in ESCs cultured in Serum/LIF, 2i/LIF, and KOSR/LIF (Appendix Fig. S1E). We found that the OCR was significantly higher for 2i/LIF and KOSR/LIF, while the ECAR was significantly lower for KOSR/LIF, and the OCR:ECAR ratio was significantly higher in ESCs cultured in 2i/LIF and KOSR/LIF compared to Serum/LIF (Appendix Fig. S1F). Moreover, "ICM-like" cells identified based on Hhex-Venus expression in 2i/LIF (Morgani et al, 2013; Riveiro and Brickman, 2020) displayed a significantly

higher OCR:ECAR ratio compared to the pluripotent fraction within the same culture (Appendix Fig. S1G,H).

We then assessed whether forcing an increased dependence on OXPHOS relative to glycolysis would be sufficient to induce an ICM-like cell state. To achieve this, we replaced D-glucose and pyruvate in Serum/LIF media with D-galactose. We refer to this media as Enhanced Metabolic Media (EMM), as it increases levels of OXPHOS and reduces dependence on glycolysis. D-galactose can only enter the glycolytic pathway due to its conversion to glucose-6-phosphate, a process requiring 2 molecules of ATP. Given that glycolysis itself only produces two molecules of ATP, glycolysis will produce no net gain of ATP and is therefore strongly inhibited (Fig. 1A, Gohil et al, 2010). As a result, the culture of ESCs in EMM should result in extensive utilization of OXPHOS to generate sufficient ATP for cell survival. Within 3 h of replacing standard ESC media with EMM, we observed a rapid simultaneous inhibition of glycolysis and stimulation of OXPHOS (Fig. 1B). This state was stabilized at 24 h, and ESCs could be maintained in this culture condition indefinitely to create Enhanced Metabolic ESCs (EMESCs). EMESCs exhibit a dramatic metabolic shift relative to ESCs, and the OCR:ECAR ratio following 24 h in EMM is considerably greater than for ESCs cultured in 2i/LIF or KOSR/LIF for three passages (Appendix Fig. S2A).

Culture of ESCs in EMM was found to promote the simultaneous and more homogeneous expression of fluorescent reporters for both the Epi/pluripotency marker Nanog and the PrE marker Hhex (Fig. 1C). To determine if EMM induces an ICM-like phenotype in EMESCs, we compared the transcriptome of EMM-cultured ESCs with mouse pre-implantation embryos from different developmental stages (Deng et al, 2014). While Serum/LIF-cultured ESCs are closer to the traditional naïve ESCs in this dataset, increased time in EMM shifted the transcriptome of EMM-cultured ESCs closer to that of the ICM (Fig. 1D). In addition, elevated levels of both NANOG protein and Hhex-Venus were present within the same cells (Appendix Fig. S2B). Pluripotency markers NANOG, OCT4, and SOX2 were also co-expressed with the PrE transcription factor (TF) GATA6 in a small population of EMESCs, suggesting these GATA6[+] cells represent an unsegregated ICM-like state rather than the spontaneous PrE differentiation observed Serum/LIF conditions (Appendix Fig. S2C). We also observed no significant increase in apoptosis in the Nanog[low] population (Appendix Fig. S2D) in response to treatment with EMM.

ESC colonies expanded in EMM were homogeneously undifferentiated (Appendix Fig. S2E), and consistent with the smaller size of EMM colonies, we observed that ESCs grown in these conditions have a considerably longer cell cycle length of about 48 h, proliferating at roughly half the rate of Serum/LIF-cultured ESCs, with a significantly increased proportion of cells in G1 phase and less in G2/M (Appendix Fig. S2F) as has been reported for 2i/LIF culture (ter Huurne et al, 2017). However, when we assessed the response of specific signaling pathways to EMM culture, it became clear that this condition has little in common with 2i/LIF; as expected based on the inclusion of the GSK3 and MEK inhibitors in 2i/LIF, levels of pERK are much higher in EMM, while active β-catenin is significantly higher in 2i/LIF (Fig. EV1A). Moreover, while pAKT follows ERK activation in EMM, pSTAT3, and pYAP levels are elevated in both EMM and 2i/LIF compared to Serum/LIF. At a transcriptional level, EMM is also different from 2i/LIF,

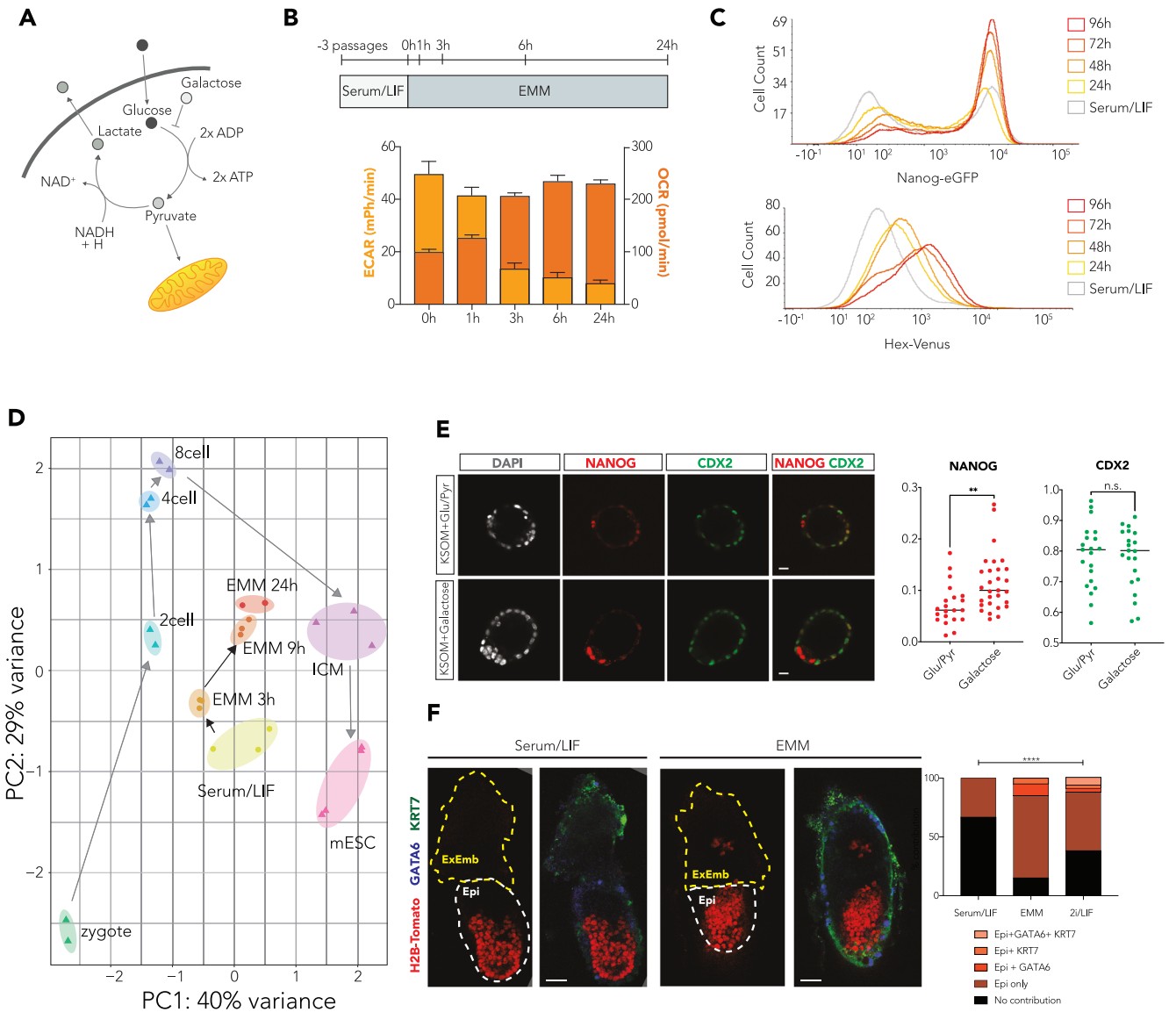

**Figure 1. Altering the metabolism of ESCs by forcing a change in the ratio of glycolysis to OXPHOS to create EMESCs increases the capacity for self-renewal and differentiation.**

(A) Schematic depicting how EMM affects metabolic activity in ESCs. (B) Basal OCR and ECAR analysis of ESCs during EMM culture at early time points up to 24 h, following three passages in Serum/LIF. $n = 24$ technical replicates, from three independent experiments. Error bars represent the standard error of the mean. (C) Flow cytometry histogram for Nanog-eGFP reporter ESCs (top) or Hhex-Venus reporter ESCs (bottom) after 1–4 d EMM culture, gated on GFP- or Venus-negative cells. Example profiles taken from $n = 4$ technical replicates, from two biologically independent samples. (D) PCA of RNA-seq from EMM-cultured ESCs compared with scRNA-seq from pre-implantation embryos (Deng et al, 2014) against a panel of 121 ICM- Epi-, PrE-, and TE-specific markers (Canham et al, 2010). (E) Confocal optical sections of immunostained embryos for CDX2 and NANOG, quantified relative to the number of DAPI-positive cells. **$P = 0.0037$, n.s. $P = 0.7847$, unpaired two-tailed $t$ test. E0.5–E5.5 mouse embryos cultured in KSOM+Glucose/Pyruvate ($n = 21$), KSOM+Galactose ($n = 29$). Scale bars: 20 μm (KSOM+Glu/Pyr) and 15 μm (KSOM +Galactose) (F) Images of chimeric mouse embryos (left) generated from morula injection with ESCs containing an H2B-Tomato reporter, previously cultured for two passages in Serum/LIF ($n = 27$), EMM ($n = 20$) or 2i/LIF ($n = 34$), Embryos were collected at E6.5 and immunostained for GATA6 and KRT7 to mark extra-embryonic endoderm and trophoblast, respectively, quantification of lineage contribution (right) was based on H2B-Tomato cells in the epiblast, or co-expressed with GATA6 and/or KRT7. ****$P < 0.0001$, two-tailed chi-square test. Scale bar: 50 μm. Source data are available online for this figure.

while pluripotency makers are generally high, the levels of endoderm (*Gata4*, *Sox7*, and *cMyc*) and trophoblast (*Gata3* and *Eomes*) are higher than either Serum/LIF or 2i/LIF (Fig. EV1B).

As with in vivo ICM cells (Grabarek et al, 2012), we found that EMESCs could differentiate into both embryonic and extra-embryonic lineages in vitro, although we observed little difference

in the ability of different ESC cultures to undergo embryonic neural differentiation (Fig. EV1C), EMESCs were found to be poised to form developmentally proximal extra-embryonic PrE (Anderson et al, 2017) (Fig. EV1D) and effectively initiated TE differentiation based on expression of *Gata3* and *Elf5* (Fig. EV1E). Over longer culture periods, EMESCs were found to express fewer apoptotic

genes than ESCs cultured in 2i/LIF after 10 and 20 passages, indicating their long-term stability, while embryonic and extra-embryonic genes continued to be expressed (Fig. EV1F–I).

To examine whether EMM culture affects embryonic development, we cultured mouse embryos from zygotes to late-stage blastocysts ex vivo, in galactose-containing media. Most embryos cultured in KSOM+galactose developed into normal blastocysts with higher numbers of NANOG-positive central cells, suggesting an expanded ICM (Fig. 1E). To determine whether EMESCs retain pluripotency, ESCs constitutively expressing a H2B-Tomato fluorescent reporter were cultured in Serum/LIF, EMM, or 2i/LIF media and then injected into wild-type morulae and analyzed at E6.5 (Fig. 1F). We found that the EMESCs were extremely efficient at generating chimeras, giving higher levels of epiblast contribution than either 2i/LIF or Serum/LIF-cultured ESCs (33% for Serum/LIF, 70% for EMM, 50% for 2i/LIF). Based on both position and whole-mount immunostaining for GATA6 and trophoblast marker KRT7, we also observed contribution to both extra-embryonic lineages at similar levels to what we observe for 2i/LIF (GATA6: 0% for Serum/LIF, 10% for EMM, 2.94% for 2i/LIF; KRT7: 0% for Serum/LIF, 5% for EMM, 2.94% for 2i/LIF) (Morgani et al, 2013). We next sought to determine whether the long-term functional capacity of EMESCs was improved over conventional ESCs, by allowing chimeric mice to develop to term, then assessing levels of chimerism and germline contribution. We found that the level of contribution of EMESCs to chimeric mice (determined by a lighter fur color, was improved by EMM culture (Fig. EV1J), including in a cell line (3.4) which normally does not give rise to chimeras with Serum/LIF-cultured ESCs. Together, these data illustrate that altering the metabolism of ESCs to a higher dependence on OXPHOS causes reprogramming of the cells to an earlier state and endows them with enhanced functional capacities.

## Metabolic change promotes ICM transcriptional states

To determine the extent of transcriptional reprogramming initiated by the immediate early shift to EMM culture, we analyzed the transcriptome of EMM-cultured ESCs at multiple time points over a 24-hour period (Appendix Fig. S3A–C). After 3 h, we observed robust transcriptional changes, with the largest of the four clusters representing downregulated genes, indicating that a major effect of EMM culture is to repress transcription (434/771 significantly differentially expressed genes, Clusters 1 + 2, Appendix Fig. S3D; Dataset EV2A). GO and KEGG pathway analysis revealed that clusters 2 and 3, representing genes down- and upregulated by EMM at 3 h, respectively, are associated with nucleosome/chromatin assembly, indicating that alteration in DNA/chromatin organization is an early downstream effect of the metabolic switch in EMM (Dataset EV2B).

The transcription of genes that characterize both the Epi and PrE lineages were largely upregulated upon EMM culture, although genes representing the 2-cell state were unchanged (Appendix Fig. S3E). While genes of the early trophoblast were also upregulated, we found that Cdx2, Elf5, and Tmem54, which characterize the differentiated trophoblast, were downregulated (Appendix Fig. S3E). When the transcriptomes of EMM-cultured ESCs were compared with those of other ESCs representing different states of pluripotency, EMM-cultured ESCs at 24 h expressed increased levels of naïve pluripotency markers, and reduced levels of primed pluripotency markers (Appendix Fig. S3F).

To determine how EMM-induced metabolic change promotes transcriptional change, we sought to define the chromatin response to the metabolic shift induced by EMM by performing time-resolved ATAC-seq at the same time points as our RNA-seq dataset (Appendix Fig. S3A,G,H). We focused on the enhancers regulating their expression (Hamilton et al, 2019) and found that for Epi-, PrE-, and TE genes (Nanog, Spry4 and Krt8, respectively) we observed an increase in the accessibility around the active enhancer regions at 24 h (Appendix Fig. S3I). While the predominant impact on gene expression was downregulation (Appendix Fig. S3D), we observed similar numbers of regions with increasing and decreasing accessibility, with a slight bias in favor of more regions opening up (Appendix Fig. S3J; Dataset EV3A). We also observed a positive correlation between RNA expression and chromatin accessibility after 24 h EMM culture (Appendix Fig. S3K; Dataset EV3B). To provide a more robust analysis of these transcriptional states, we defined new enhancer sets involved in regulating these cell states using CUT&Tag for histone marks associated with active enhancers, H3K27ac and H3K4me1 (Fig. EV2A-F). Consistent with the overall reduction in transcription levels we observed a reduction in the number of active enhancers, from 13,705 to 3829 (Fig. EV2A), but at enhancers where the chromatin opened, we observed an increased level of acetylation, suggesting higher levels of activity (Figs. 3A and EV2F, top); a trend reflected in the opening of the shared enhancer set over time in EMM (Fig. EV2C). Conversely, we also noted a reduction of acetylation at enhancers where the chromatin is closing (Figs. 3A and EV2C, F, bottom).

Common motifs observed in open EMM enhancers include TEAD, AP1, and JAK/STAT, possibly reflecting the activation of MAPK, Hippo and LIF signaling in these cells and the presence of these GO terms for LIF, placenta, and blastocyst formation in the EMM enhancer set (Fig. EV2B,D). While motifs for the pluripotency factors OCT4 and SOX2 are better represented in closing enhancers, KLFs and ESRRB are strongly represented in EMM opening enhancers (Fig. EV2D).

We also explored regulation around the complete set of Serum/LIF-EMM shared enhancers (Fig. EV2E). As with the EMM enhancer set, ESC enhancers that respond to EMM regulate genes associated with extra-embryonic development and an assortment of metabolic responses (Fig. EV2G, top; Dataset EV3C), whereas enhancers that become less accessible at 24 h EMM culture were associated with biological processes that occur later in development (Fig. EV2G, bottom; Dataset EV3D). Collectively, EMM shuts down a large set of potentially spurious enhancers in ESCs and reinforces the activity of those associated with pre-implantation development and the signaling pathways that regulate it.

## Metabolic analysis of EMM culture reveals a role for NAD⁺-dependent Sirtuin activity

To connect the transcriptional response observed in EMM-cultured ESCs to a metabolic change, we quantified the relative abundance of metabolites in ESCs cultured in either Serum/LIF or EMM at multiple time points over a 24-h period (Fig. EV3A–C). We identified 85 metabolites that were differentially expressed between the two media conditions at all time points (Fig. EV3D; Dataset EV4). At the 3 h time point, we observed induction of L-Carnitine, with a reciprocal reduction in L-Acetylcarnitine (Fig. EV3E), indicating that ESCs are rapidly adapting to glucose depletion and

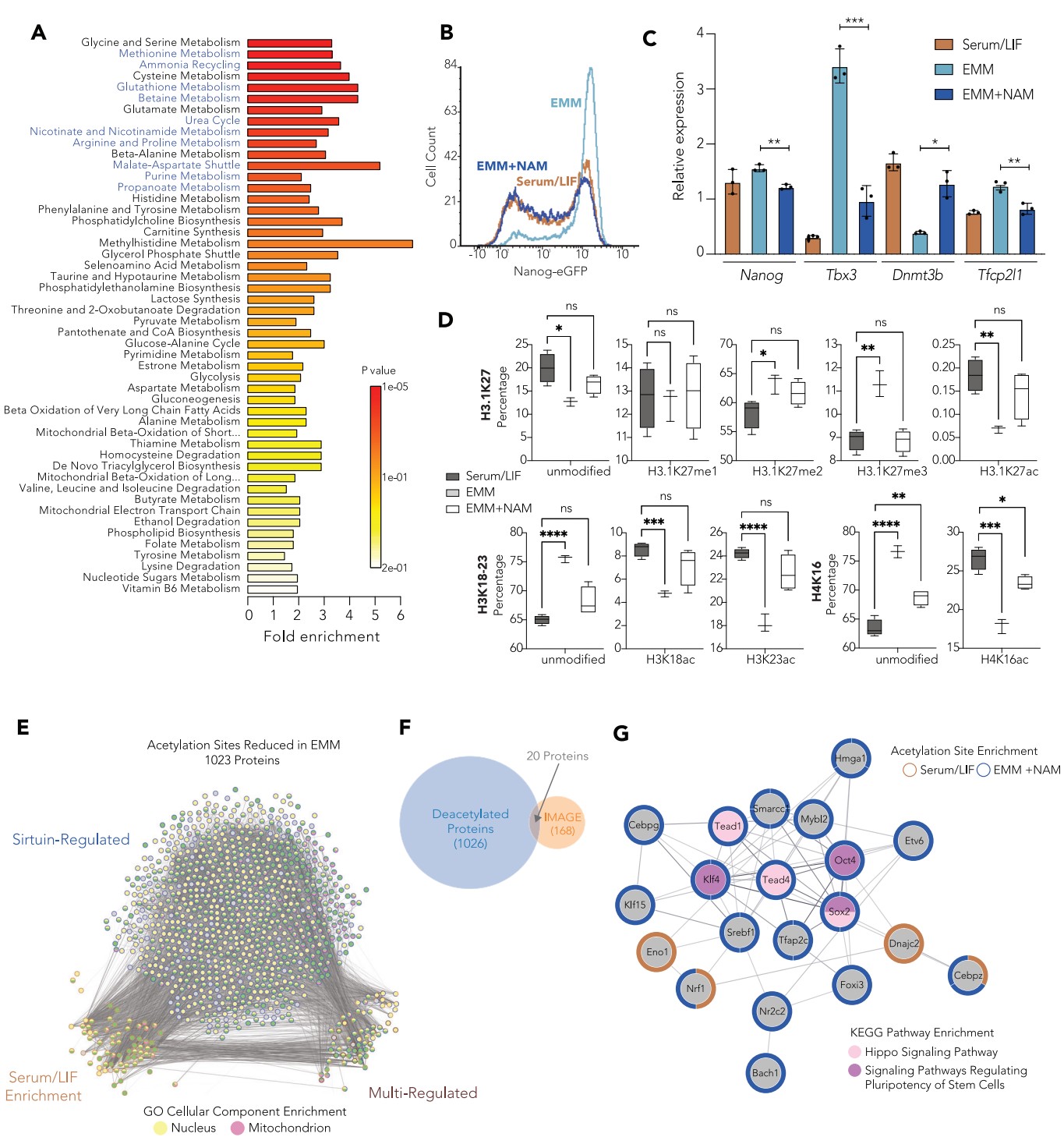

catabolizing other molecules, such as fatty acids and amino acids, in order to generate intermediate metabolites for the tricarboxylic acid (TCA) cycle. Consistent with the need to engage fatty acid metabolism via the TCA cycle, we found that culturing ESCs in EMM elevated levels of the mitochondrial membrane fatty acid transporter CPT1A by immunofluorescence (Fig. EV3F) and in our EMM RNA-seq dataset (Fig. EV3G). Fatty acid oxidation,

measured by 14C-Palmitate oxidation, was also increased in EMM (Fig. EV3H).

Pathway enrichment analysis of the differentially regulated metabolites revealed that in addition to alterations in individual amino acid metabolism, "Nicotinate and Nicotinamide Metabolism" was highly enriched (Fig. 2A). These factors are the components of the NAD+ salvage pathway (Fig. EV3I), which is

**Figure 2.  Metabolic changes driven by EMM culture and their impact on EMESC phenotypes.**

(A) Metabolite Set Enrichment Analysis (MSEA) of metabolic pathways that are differentially expressed during EMM culture, as analyzed by MetaboAnalyst 2.0 (see "Methods"). (B) Flow cytometry histogram for Nanog-eGFP ESCs cultured in Serum/LIF and EMM −/ + NAM for two passages. Data are representative of three experimental replicates. (C) RT-qPCR analysis of EMM immediate response genes in ESCs cultured in Serum/LIF and EMM − / + NAM for 24 h. Data are mean + s.d., $n = 3$ biologically independent samples. P values: Nanog EMM vs EMM + NAM **$P = 0.0093$; Tbx3 EMM vs EMM + NAM ***$P = 0.0005$; Dnmt3b EMM vs EMM + NAM *$P = 0.0164$; Tcfp2l1 EMM vs EMM + NAM **$P = 0.0048$. Two-tailed unpaired t test. (D) Global histone PTM levels quantified by mass spectrometry. $n = 4$ biological replicates (3 for EMM). Lines indicate median, boxes represent first and third quartiles and whiskers extend 1.5× IQR. P values: unmodified H3K27: *$P = 0.0136$, ns $P = 0.1225$; H3K27me1: ns (left) $P = 0.7916$, ns (right) $P = 0.8986$; H3K27me2: *$P = 0.0297$, ns $P = 0.0851$; H3K27me3: **$P = 0.0016$, ns $P = 0.8683$; H3K27ac: **$P = 0.0029$, ns $P = 0.2464$; unmodified H3K18K23: ****$P < 0.001$, ns $P = 0.559$; H3K18ac: ***$P = 0.0002$, ns $P = 0.1430$; H3K23ac: ****$P < 0.001$, ns $P = 0887$; unmodified H4K16: ****$P < 0.001$, **$P = 0.0021$; H4K16ac: ***$P = 0.0003$, *$P = 0.0105$. Two-tailed unpaired t test. (E) Node STRING map depicting 1023 (out of 1026 queried) proteins deacetylated in EMM relative to Serum/LIF and/or EMM + NAM (Sirtuin-regulated). (F) Venn diagram depicting overlap between TFs enriched in IMAGE analysis (168) and proteins that were significantly deacetylated in EMM (1023). (G) STRING diagram and KEGG Pathway analysis of 20 TFs enriched in acetylome and IMAGE analysis. Source data are available online for this figure.

essential for many metabolic pathways throughout the cell, including the regulation of chromatin-associated proteins involved in gene transcription (Berger and Sassone-Corsi, 2016). In addition, we also identified 10/14 of the top significantly enriched metabolic pathways (Fig. 2A, blue font) as associated with NAD⁺ homeostasis. There are two main classes of NAD⁺-dependent nuclear enzymes: Sirtuins and PARPs, which often regulate each other's activity, and are thought to compete for NAD⁺ (Bai et al, 2011; Pillai et al, 2005; Zhang et al, 2012). We therefore asked whether either of these enzymes were responsible for mediating the transcriptional effects of metabolic reprogramming induced by EMM. We found that PARylation activity of PARP enzymes was drastically reduced in EMM culture after 24 h (Appendix Fig. S4A) and that PARP inhibitors could not block EMESC induction by EMM (Appendix Fig. S4B), suggesting that the immediate early impact of NAD⁺ on transcription is likely due to Sirtuin activity.

Sirtuin-mediated deacetylation of histones and TFs has been linked to both somatic and ESC self-renewal, differentiation, and reprogramming (Ryall et al, 2015; Williams et al, 2016). Sirt1, in particular, has been shown to be both the most highly expressed Sirtuin in the mouse and human (Nowotschin et al, 2019; Li et al, 2022; Proks et al, 2025) pre-implantation embryo, and has been shown to act in reprogramming to induced pluripotent cells (Lee et al, 2012) and the maintenance of pluripotency (Williams et al, 2016). We found that the transition of ESCs to ICM-like EMESCs induced by EMM was inhibited by NAM, which acts as a potent SIRT inhibitor (Bitterman et al, 2002) (Fig. 2B,C). Based on histone-targeted mass spectrometry, we observed that EMM-cultured ESCs have reduced acetylation of histone lysines at H3K9, H3K27 and H4K16 after 24 h (Fig. 2D; Appendix Fig. S4C,D; Dataset EV5). We also noted a general decrease of histone acetylation alongside a reciprocal gain in methylation (Appendix Fig. S4E). The addition of NAM also blocked the capacity of EMM to promote the co-expression of NANOG within the GATA6-positive cells in EMM culture (Appendix Fig. S4F).

We found that the SIRT1 inhibitor Ex-527 (Gertz et al., 2013) partially blocked EMM-mediated increases in Nanog expression, suggesting that additional members of the Sirtuin family could be involved in the metabolic induction of ICM-like transcription (Appendix Fig. S4G). To rule out the possibility that EMM acts through general levels of histone acetylation, we used an inhibitor for the broad-acting histone acetylases CREBBP (CBP) and EP300 (A-485 Lasko et al, 2017). We found that transcription of Nanog was reduced to a similar degree in both Serum/LIF- and EMM-

cultured ESCs after 3 h (Appendix Fig. S4H,I), indicating that a specific pattern of deacetylation regulated by Sirtuin proteins is responsible for the EMESC phenotype.

## Deacetylation of ICM-specific TFs mediates enhancer occupancy in EMESCs

We sought to identify Sirtuin-dependent deacetylations in EMM by performing an acetylome on ESCs cultured in Serum/LIF, EMM or EMM + NAM for 24 h, before immunopurification of acetyl-lysine, followed by quantitative LC-MS/MS (Fig. EV4A). As expected, the number of acetylated lysines was higher in EMM + NAM compared to Serum/LIF and EMM, since NAM is a potent inhibitor of Sirtuin deacetylases (Fig. EV4B–D). We then identified the sites that were specifically deacetylated in EMM relative to the other two conditions in response to this fundamental shift in metabolism (Figs. 2E and EV4E; Dataset EV6A). In EMM 1871 sites were deacetylated out of a total number of 6733 identified sites (Fig. EV4F,G; Dataset EV6A,B). Moreover, most of these proteins were enriched in the nucleus (77%), with 12% enriched in chromatin, indicating that the major response to the acetylome occurs in the nucleus and therefore affects nuclear processes such as transcription (Fig. EV4F,G; Dataset EV6A,B).

Acetylation is generally associated with enhancer activation, and we observed extensive acetylation of CBP and EP300 across all conditions (Dataset EV7). However, in both proteins none of the commonly identified acetylated sites in the KAT domain (Thompson et al, 2004; Wang et al, 2005) were deacetylated by Sirtuins. While we detected several Sirtuin-regulated deacetylations in CBP/EP300, they were outside of defined regulatory domains and have no known effect on catalytic function (Dataset EV7), consistent with our observation that inhibition of the KAT activity of EP300/CBP does not produce a similar phenotype to EMM (Appendix Fig. S4H,I). We next compared our dataset to previously reported EP300/CBP-dependent mouse embryonic acetylomes (Weinert et al, 2018; Narita et al, 2021). We observed considerable overlap in specific acetylated residues (Fig. EV4G; Dataset EV8A,B) and therefore assessed whether sites identified as Sirtuin-dependent were also regulated by EP300/CBP. We found that 6% of all detected acetylations were EP300/CBP-regulated (comparable to the levels of EP300/CBP regulation reported in Weinert et al, 2018; Narita et al, 2021), with only a minor enrichment of EP300/CBP-regulated sites that were also Sirtuin-dependent (Dataset EV8C–E). As EP300/CBP and Sirtuins are chromatin regulators, we focused

on whether this minor enrichment is merely a byproduct of transcription and constructed a network of Sirtuin-regulated proteins associated with chromatin regulation, transcriptional activity, and stem cell maintenance (Dataset EV9). Based on the overlap of this network with sites known to be regulated by EP300/CBP, there was no enrichment of proteins with co-regulated acetylation sites. In addition to these factors, we identified acetylation sites in canonical pluripotency and ICM-related TFs that were Sirtuin-, but not EP300/CBP-regulated (e.g., SOX2, OCT4, NANOG, TEAD4, YAP1, TFAP2C, MYC; Dataset EV9).

To deduce which primary TFs within our acetylome are responsible for metabolic regulation of transcription, we used the algorithm, "Integrated analysis of Motif Activity and Gene Expression" (IMAGE) to decipher the functional elements within regulatory regions and predict TFs that are likely to be causal in inducing changes in gene expression (Grud et al, 2018). We applied IMAGE to our RNA- and ATAC-seq data across the time course for EMM induction and predicted 168 causal TFs (Dataset EV10A). KEGG pathway analysis revealed an enrichment of TFs involved with cancer, general pluripotency, and the Hippo signaling pathway. Of this set of potential causal TFs identified by IMAGE, 20 were also regulated at the level of acetylation (Fig. 2F, Dataset EV10B). This reduced list maintains KEGG pathway analysis enrichment for pluripotency and the Hippo pathways (Fig. 2G, Dataset EV10C), suggesting that the acetylation of these factors could determine the transcriptional state adopted by EMESCs.

As histone deacetylation dampens enhancer activity, we assume one or all of these TFs drive selective enhancer activation. Thus, we asked whether there was a specific TF that was deacetylated in a position that would be predicted to enhance its affinity for DNA. From the list of 20 TFs, only one has lysines that are acetylated in its DNA binding domain, SOX2 (Fig. 2G, Dataset EV10D). We had also observed that SOX2 acetylation is increased in EMM, but this acetylation is lost with the addition of NAM, (Fig. EV4H). We therefore asked how EMM influences SOX2 binding and whether this correlates with the recruitment of H3K27ac, or the recruitment of two other TFs that are deacetylated, but not in their DNA binding domain, TEAD1 and KLF4. CUT&Tag for these factors revealed that they are all found clustered with increased levels of acetylation at regulated enhancers (Fig. 3B,C), and that SOX2 binding to both open and closed enhancer regions was three- to fourfold stimulated by EMM (Fig. 3C). While the binding of both TEAD1 and KLF4 is globally reduced in EMM, their binding is enhanced at regions where SOX2 is bound, particularly in opening regions (Fig. 3C). Motif analysis of the new SOX2 peaks in EMM shows significant enrichment for these deacetylated TFs (Fig. 3D). Taken together, this suggests that SIRT1-mediated deacetylation of these factors drives their cooperative binding to sites that exhibit enhanced SOX2 binding in EMM. In this way, deacetylated histones would suppress transcriptional or epigenetic noise at the same time as the deacetylated TF drives specific enhancer activation.

### SIRT1 deacetylation of SOX2 drives cooperative response to metabolic change

To probe the generality of this mechanism, we began by asking whether simultaneously enhancing histone and TF affinity would enhance specificity with a simple mass action model. We invoked cooperativity in the form of a Hill coefficient and showed a profound change in productive binding in response to simultaneously increasing both affinities (Fig. 3E). We next thought to validate the role of SIRT1 in this process by generating a conditional Sirt1 knockdown ESC line. Based on the removal of the catalytic domain at exons 4-6 (Appendix Fig. S5A), we both eliminate SIRT1 protein and the ability of EMM to promote histone deacetylation (Appendix Fig. S5B–D), indicating that SIRT1 is likely responsible for histone deacetylation in response to EMM. Consistent with previous observations in standard ESC conditions (Baltus et al, 2009), we also observed that SIRT1 was required to deacetylate SOX2 in EMM (Fig. EV4H; Appendix Fig. S5E).

Having shown that SIRT1 is required to deacetylate SOX2 in EMM, we sought to demonstrate the specific influence of SIRT1 on SOX2 occupancy of chromatin in EMM. To do this, we rescued Sirt1-KO ESCs with a SIRT1-FKBP fusion under the control of tetracycline (Appendix Fig. S5F–H), in which a homozygous knockout mutant was rescued by SIRT1-FKBP that can be rapidly degraded by the addition of the molecule dTag-13. Using these mutants, we demonstrate that histone deacetylation in response to EMM is reduced when SIRT1 is rapidly degraded (Appendix Fig. S5I). In parallel, we found that the majority of EMM-stimulated SOX2 binding at enhancers is SIRT1-dependent, and this is most pronounced where the ATAC signal is weakest (Figs. 3F–H and EV5A,B).

Finally, in order to assess whether deacetylation of SOX2 in its HMG domain is sufficient to drive the EMM response, we generated SOX2 constructs with the three potential SIRT1 target lysines identified in the HMG domain mutated to either arginine (mimicking deacetylated lysine, 3K-R) or glutamine (mimicking acetylated lysine, 3K-Q), and assessed their capacity to generate ESC colonies in EMM in a SOX2-FKBP ESC line (Liu et al, 2021) (Fig. 4A,B). Two of these residues (K67 and K97) were identified in our acetylome (Fig. 2G), and one (K75) was previously identified as a SIRT1-regulated lysine (Mu et al, 2015). Rescue of SOX2 expression with the K-R construct modestly enhances the capacity of SOX2-depleted ESCs to generate naïve colonies, whereas the K-Q construct totally blocks the ability of EMM to induce its enhanced self-renewal phenotype and generate naïve colonies (Fig. 4C). These data suggest that a general metabolic change can elicit a specific phenotype by the simultaneous deacetylation of chromatin alongside a sequence-specific TF.

## Discussion

We found that ESCs can be reprogrammed metabolically to a more ICM-like state by forcing enhanced dependence on OXPHOS at the expense of glycolysis. The induction of ICM-like EMESCs was rapid and exploited alterations in $NAD^+$ metabolism derived from this metabolic switch, that drive deacetylation of essential TFs. In general, this enhanced specificity is achieved by simultaneously increasing the affinity of the TF and that of chromatin, to reduce epigenetic or transcriptional noise at the same time as enhancing a specific transcriptional response. Based on our work with the pluripotency network, we find this likely involves a trigger TF with deacetylation in its DNA binding domain, coupled to cooperativity with a suite of other deacetylated TFs.

Increased levels of $NAD^+$ turnover appear to be responsible for the transcriptional response observed in EMESCs, and lineage-specific

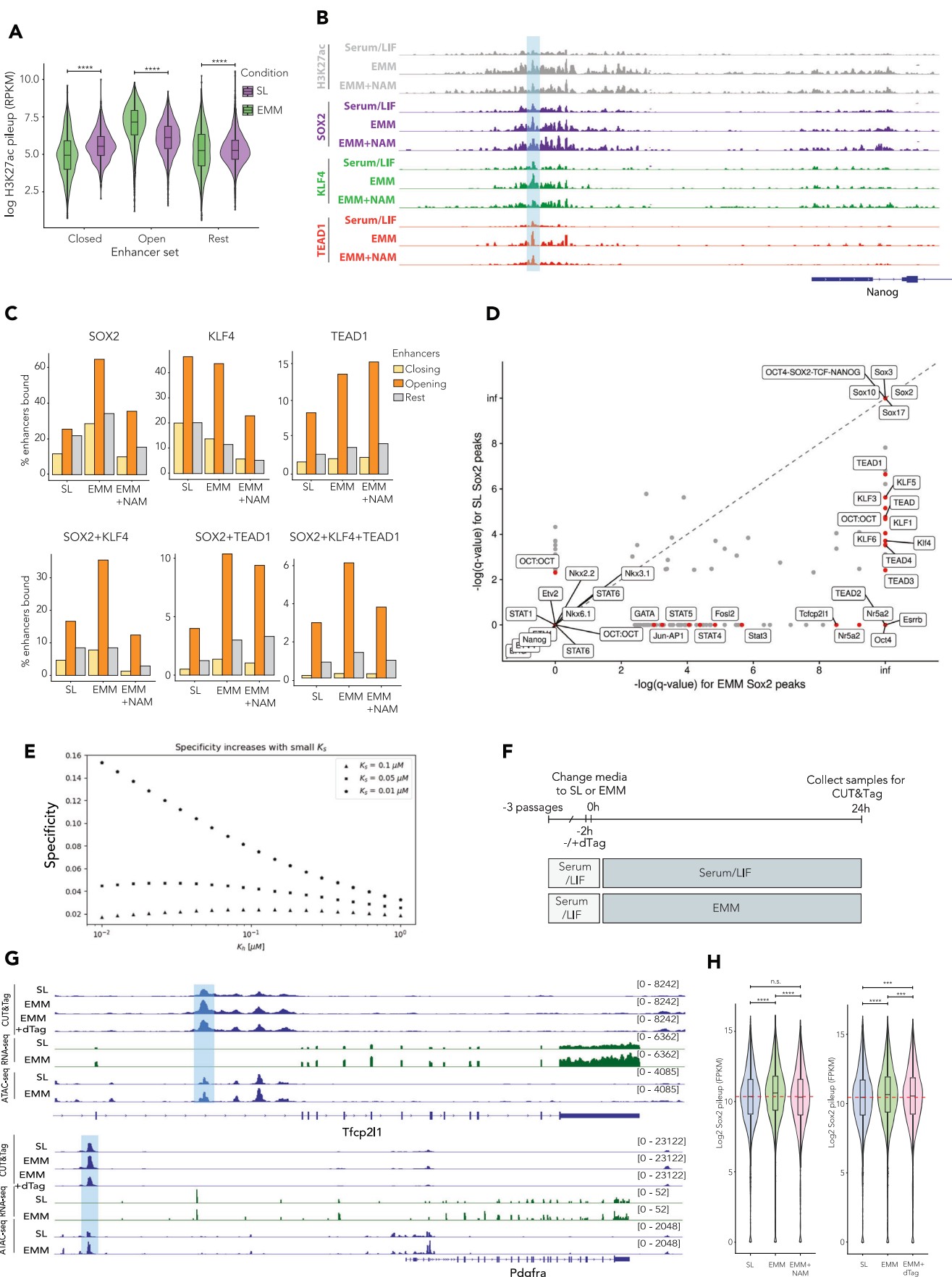

**Figure 3.  Sirtuin-dependent deacetylation of lineage-specific transcriptional regulators underlies EMM phenotype.**

(A) Violin plot depicting H3K27ac distribution at opening, closing, and non-changing enhancers in SL and EMM conditions. Peaks present in 3/3 replicates per condition for H3K27ac, ****$P < 0.0001$, one-way ANOVA test. Lines indicate median, boxes represent first and third quartiles and whiskers extend 1.5× IQR. Bonferroni-corrected $P$ values: SL Closed vs EMM Closed ****$P < 0.001$; SL Open vs EMM Open ****$P < 0.001$; SL non-changing vs EMM non-changing ****$P < 0.001$. Wilcoxon rank-sum test. Lines indicate median, boxes represent first and third quartiles and whiskers extend 1.5× IQR. (B) Genome browser track (IGV 2.14.0 software of an example region showing the distribution of H3K27ac, as well as the transcription factors SOX2, KLF4 and TEAD1 at the Nanog proximal enhancer. (C) Percent of enhancers in each set overlapped by peaks co-bound by SOX2, KLF4, and TEAD1. Peaks present in 2/3 replicates per condition for KLF4 and TEAD1, and in 4/6 reps for SOX2. (D) Motif enrichment analysis on SL vs EMM peaks for SOX2 using 4/6 biological replicates over two experiments. (E) Mathematical model depicting the increase in specificity, e.g., SOX2 binding to pluripotency enhancers with increasing binding affinity of SOX2 to DNA versus histone binding affinity to DNA. (F) Experimental outline depicting SOX2 CUT&Tag assay in Sirt1-FKBP cell line. (G) Genome browser tracks depicting SOX2 binding in the Sirt1-FKBP ES cell line, in SL, EMM and EMM+dTag-13 conditions, with RNA-seq and ATAC-seq tracks at the *Tfcp2l1* (top) and *Pdgfra* (bottom) loci. Blue boxes depict opening enhancers. (H) Violin plot showing SOX2 binding (log2 read count) distribution at all SL-EMM enhancers in WT ESCs. In SL, EMM and EMM + NAM conditions. Peaks present in 3/3 replicates per condition for SOX2. Bonferroni-corrected $P$ values: WT SL vs EMM ****$P < 0.001$; WT EMM vs EMM + NAM ****$P < 0.001$; WT SL vs EMM + NAM n.s. $P = 0.11$; Sirt1-dTag SL vs EMM ****$P < 0.001$; Sirt1-dTag EMM vs EMM+dTag ***$P = 0.00033$; Sirt1-dTag SL vs EMM+dTag ***$P = 0.0002$. Wilcoxon rank-sum test. Lines indicate median, boxes represent first and third quartiles and whiskers extend 1.5× IQR.

responses to alterations in the ratio of OXPHOS to glycolysis. This is reminiscent of observations that suggest NAD$^+$ recycling is linked with increased levels of stem cell potency and reduced cell aging in a wide variety of contexts (Calvanese et al, 2010; Son et al, 2016; Liu et al, 2019). Sirtuins and PARPs, the two major groups of NAD$^+$-consuming nuclear enzymes, appear to be disproportionally affected by EMM culture, with PARP activity reduced as Sirtuin-mediated deacetylation increases. This could reflect the fact that Sirtuins are more sensitive to fluctuations in local NAD$^+$ levels than PARPs (Houtkooper et al, 2010; Mendoza-Alvarez and Alvarez-Gonzalez, 1993), making these deacetylases the logical drivers of NAD$^+$ mediated cellular phenotypes. In particular, Sirtuins are known to have regenerative properties in a variety of species (Bonkowski and Sinclair, 2016), and perhaps their capacity to activate early embryonic programs is central to some of these phenomena. In particular, the molecular mechanism we describe here (see Model, Fig. 4D), in which reduction of transcriptional noise occurs with the elevation of specific signals, could underlie how these factors attenuate the aging process. Thus, this family of chromatin modifiers could enhance lineage-specific signatures while reducing non-specific transcription that leads to the phenotype variation produced in response to aging.

While the acquisition of an EMESC ICM-like state relied on "activating" deacetylation of key TFs, it was also accompanied by an apparent reduction in histone acetylation. This is consistent with the observation that the primary response to EMM is transcriptional inhibition. How then does the activation of Sirtuin deacetylases lead to the observed increases in accessibility at ICM-like enhancers and accompanying changes in gene expression, despite these reductions in histone acetylation? We detected robust acetylation of EP300 and CBP in all conditions and the treatment of ESCs with a KAT inhibitor did not phenocopy EMM, but rather produced the same global inhibition of transcription that had been observed previously (Lasko et al, 2017; Weinert et al, 2018). Thus, Sirtuin activity may result in reduced histone, but not regulatory EP300/CBP acetylation, suggesting deacetylated TFs can directly recruit EP300/CBP to specific enhancers. As a result, we observed that enhancer sequences marked by H3K27ac exhibit enhanced accessibility in EMM, despite the overall reduction in H3K27ac levels. That the dominant lineage-specific transcriptional output of the Sirtuin family in response to metabolic reprogramming occurs via the deacetylation of TFs associated with ICM identity might explain why these enzymes have been linked to the acquisition of

pluripotency (Staszkiewicz et al, 2013; Williams et al, 2016; Kim et al, 2018). Moreover, we observe two classes of Sirtuin-responsive TFs (Fig. 4D), in the instance of SOX2, the deacetylation occurs in the DNA binding domain and has global effect on SOX2 binding. For the other class of TFs, the deacetylation occurs elsewhere and appears to drive enhanced occupancy, but in regions where SOX2 is also present.

While ESCs in states that resemble earlier stages of development, such as in 2i/LIF, are characterized by higher levels of histone acetylation (Marks and Stunnenberg, 2014), the opposite is observed in EMESCs. Yet, despite these reduced levels of histone acetylation, EMESCs exhibit enhanced contribution to chimeras and extra-embryonic differentiation. Does this mean that EMESCs have a less noisy and higher fidelity steady state than ground-state ESCs? Perhaps OXPHOS establishes a ground state for all development, while the enhanced dependence on glycolysis in Epi, PrE and TE restricts lineage choice. Around implantation, cells in the ICM are reliant upon OXPHOS, which is subsequently downregulated as cells transit into peri-implantation epiblast, and glycolysis is upregulated (Shyh-Chang et al, 2013). Similarly, TE cells also have higher rates of glucose metabolism than ICM cells (Houghton, 2006). When embryos are cultured in reduced nutrient conditions, the addition of pyruvate increases the number of TE cells (Ermisch et al, 2020), consistent with our observation that the number of ICM cells are increased in embryos deprived of glucose metabolism.

Although initial inspection of the TFs deacetylated in EMESCs reveals a number of naïve pluripotency factors, these are all expressed in the ICM and a number of them are also expressed in the early PrE in addition to the Epi (Morgani and Brickman, 2015; Frum et al, 2013; Wicklow et al, 2014; Nowotschin et al, 2019, Linneberg-Aggerholm et al, 2024). The culture of embryos in EMM appears to generate a development stasis, in which there are more NANOG-positive cells in a manner that is reminiscent to diapause. In this respect, it is intriguing that a recently published report links the principal EMM component, galactose to human diapause (Iyer et al, 2024).

To link an ubiquitous cellular process such as energy metabolism to cell programming requires an intricate set of metabolically coupled enzymes that can act directly on transcriptional programs. Here, we identified a way in which a binary metabolic switch could program a complex change in cell states and potentially links metabolic fitness to lineage choice. Thus, an

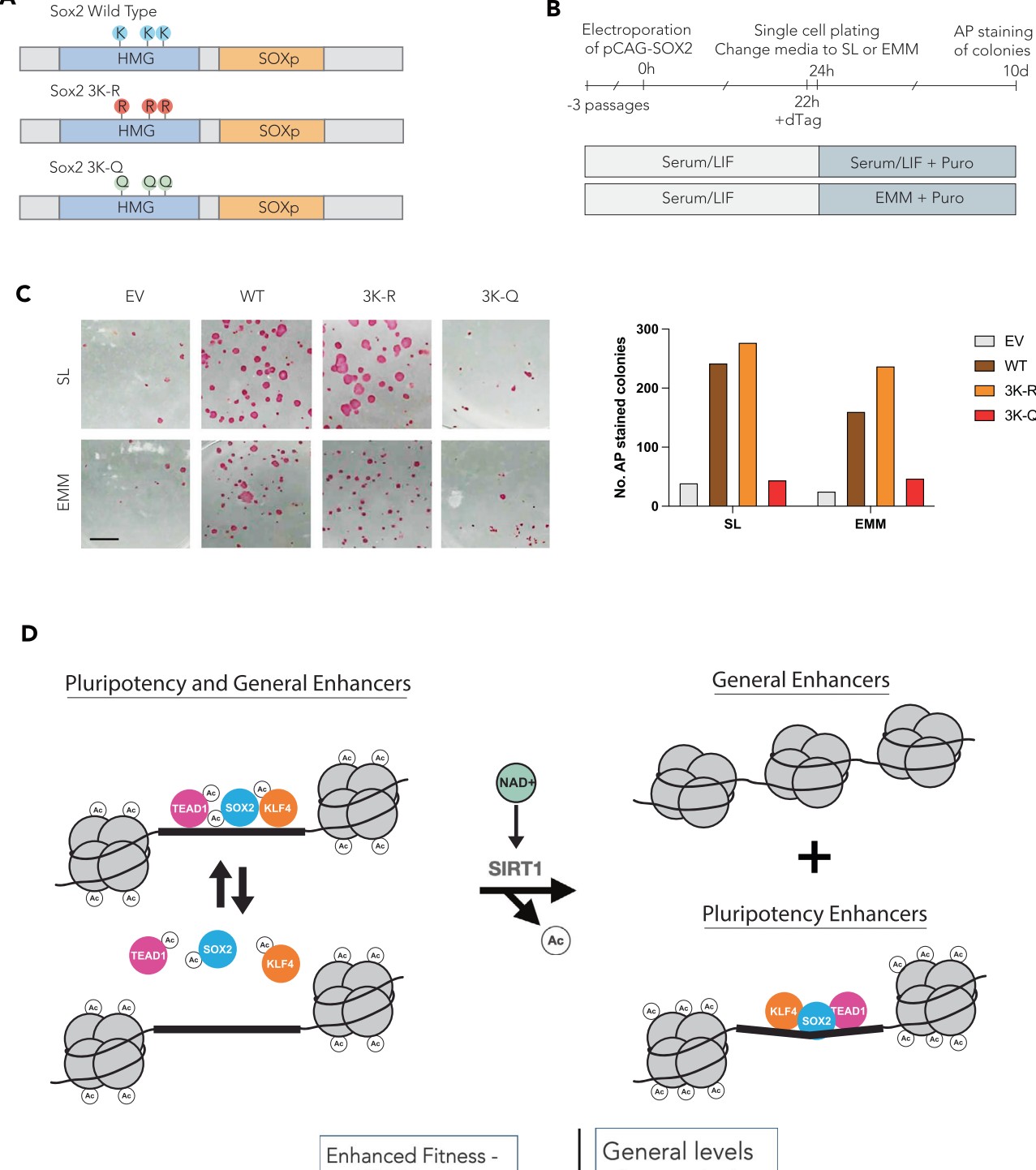

**Figure 4. Specific deacetylation of SOX2 by SIRT1 is required for EMM phenotype.**

(A) Schematics of SOX2 WT, 3K-R (deacetylation phenotype), and 3K-Q (acetylation phenotype). (B) Experimental outline depicting colony assay formation in SOX2-FKBP cell line treated with dTag-13 and transfected SOX2 transgenes. (C) Images of Alkaline Phosphatase-stained colonies after 9 days culture in SL or EMM with Puromycin selection, with quantification of AP-stained colonies from two experiments (right). Scale bar = 50 mm. (D) Model depicting how activated SIRT1 regulates transcription factors and histones to promote an ICM-like (pluripotent) identity, downstream of a metabolic shift induced by EMM culture. Source data are available online for this figure.

enhanced dependence on OXPHOS characterized here by higher levels of fatty acid oxidation and the suppression of glycolysis induces an ICM-like foundation state for lineage specification. This implies that reducing metabolic or transcriptional noise enhances differentiation competence and that this metabolically "quiet" state was selected to have the greatest developmental potential. Moreover, the decay of this transcriptional signal-to-noise ratio could be an essential component of aging (Bartz et al, 2023), as progenitor or stem cells produce a spectrum of unwanted phenotypes in response to reducing this ratio. Consequently, this phenomenon and the ability of $NAD^+$-coupled Sirtuin activity to counteract it, could underlie the reduced fitness of stem cell populations that occurs with aging and the ability of these enzymes to act on this process.

# Methods

### Reagents and tools table

| Reagent/resource | Reference or source | Identifier or catalog number |
|---|---|---|
| **Experimental models** | | |
| E14tg2a Wild-Type ESCs (Mus musculus) | Gift from Kunath lab, University of Edinburgh | N/A |
| Hex-Venus reporter ESCs (Mus musculus) | Canham et al, 2010 | N/A |
| H2B-Tomato reporter ESCs (Mus musculus) | This study | N/A |
| Nanog-EGFP reporter ESCs (Mus musculus) | Gift from Thompson lab, USCF | N/A |
| Sirt1-GFP-FKBP ESCs (Mus musculus) | This study | N/A |
| Sox2-mCherry-FKBP ESCs (Mus musculus) | Gift from De Wit lab, NCI, Amsterdam | N/A |
| **Recombinant DNA** | | |
| pTET-Sirt1-FKBP-GFP | This study | N/A |
| pCAG-FLAG.SOX2 WT | This study | N/A |
| pCAG-FLAG.SOX2 3K-R | This study | N/A |
| pCAG-FLAG.SOX2 3K-Q | This study | N/A |
| **Antibodies** | | |
| Rat anti-NANOG | eBioscience | 14-5761 |
| Rabbit anti-GFP conjugated to Alexa Fluor 488 | Thermo Scientific | A-21311 |
| Goat anti-GATA6 | R&D | AF1700 |
| Mouse anti-CYTOKERATIN7 | Santa Cruz | sc70936 |

| Reagent/resource | Reference or source | Identifier or catalog number |
|---|---|---|
| Mouse anti-H4K16ac | Thermo Scientific | MA5-27794 |
| Rabbit anti-H3K9ac | Abcam | ab10812 |
| Rabbit anti-H3K27ac | Abcam | ab4729 |
| Rabbit anti-HISTONE H3.3 | Merck | cs207327 |
| Goat anti-SOX2 | R&D | RDP320-025 |
| Goat anti-OCT3/4 | Santa Cruz | sc8629 |
| Mouse anti-CDX2 | BioGenex | #MU392A-UC |
| ANNEXIN V-FITC conjugated | Sigma-Aldrich | A13199 |
| Mouse anti-CPT1A | Abcam | ab128568 |
| Goat anti-KLF4 | R&D | AF3158 |
| Rabbit anti-H3K4me1 | Abcam | ab8895 |
| Rabbit anti-TEAD1 | CST | D9X2L |
| Rabbit anti-phospho-ERK1/2 | CST | 4370 |
| Mouse anti-ERK1/2 | CST | 9107 |
| Mouse anti-Active B-CATENIN (anti-ABC) | Millipore | 05-665 |
| Rabbit anti-phospho-AKT (Ser473) | CST | 4060 |
| Mouse anti-AKT | CST | 2966 |
| Rabbit anti-phospho-STAT3 (Tyr705) XP | CST | 9145 |
| Mouse anti-STAT3 | CST | 9139 |
| Rabbit Poly/Mono-ADP Ribose (E6F6A) | CST | 83732 |
| **Oligonucleotides and other sequence-based reagents** | | |
| **CRISPR guides** | | |
| Sirt1-KO CRISPR guide1 | fw oligo: CACCGGTCTGGGAAGTCCACCGCA | N/A |
| | rv oligo: AAACTGCGGTGGACTTCCCAGACC | N/A |
| Sirt1-KO CRISPR guide2 | fw oligo: CACCGATAAAGAGTTCTCACGTGT | N/A |
| | rv oligo: AAACACACGTGAGAACTCTTTATC | N/A |
| **qPCR primers** | | |
| *Nanog* | fw oligo: CCTCCAGCAGATGCAAGAA | N/A |
| | rv oligo: GCTTGCACTTCATCCTTTGG | N/A |
| *Tbx3* | fw oligo: TTGCAAAGGGTTTTCGAGAC | N/A |
| | rv oligo: TGCAGTGTGAGCTGCTTTCT | N/A |

| Reagent/resource | Reference or source | Identifier or catalog number |
|---|---|---|
| Dnmt3b | fw oligo: TGAATGACAAGAAAGACATCTCAAG | N/A |
| | rv oligo: CGGGTAGGTTACCCCAGAAG | N/A |
| Tfcp2l1 | fw oligo: CCAAGTCTCCTCTTTTGGATTTC | N/A |
| | rv oligo: CCTAGAGGAGTTCTGAACATACATGA | N/A |
| Oct4 | fw oligo: GTTGGAGAAGGTGGAACCAA | N/A |
| | rv oligo: CTCCTTCTGCAGGGCTTTC | N/A |
| cMyc | fw oligo: CCTAGTGCTGCATGAGGAGA | N/A |
| | rv oligo: TCTTCCTCATCTTCTTGCTCTTC | N/A |
| Gata4 | fw oligo: TTCGCTGTTTCTCCCTCAAG | N/A |
| | rv oligo: CAATGTTAACGGGTTGTGGA | N/A |
| Sox7 | fw oligo: GCGGAGCTCAGCAAGATG | N/A |
| | rv oligo: GGGTCTCTTCTGGGACAGTG | N/A |
| Eomes | fw oligo: ACCGGCACCAAACTGAGA | N/A |
| | rv oligo: AAGCTCAAGAAAGGAAACATGC | N/A |
| Gata3 | fw oligo: TTATCAAGCCCAAGCGAAG | N/A |
| | rv oligo: TGGTGGTGGTCTGACAGTTC | N/A |
| Nestin | fw oligo: CTGCAGGCCACTGAAAAGT | N/A |
| | rv oligo: TTCCAGGATCTGAGCGATCT | N/A |
| Zfp521 | fw oligo: GAAACCGAGATCCCTCAAAGA | N/A |
| | rv oligo: TTCTGGCCTCTTCTTGCAGT | N/A |
| Gata6 | fw oligo: GGTCTCTACAGCAAGATGAATGG | N/A |
| | rv oligo: TGGCACAGGACAGTCCAAG | N/A |
| Pdgfra | fw oligo: GTCGTTGACCTGCAGTGGA | N/A |
| | rv oligo: CCAGCATGGTGATACCTTTGT | N/A |
| Elf5 | fw oligo: GCAGCTCTGCAGCATGAC | N/A |
| | rv oligo: TTCAGCATCATTGAAAAAGGAG | N/A |
| **Chemicals, enzymes, and other reagents** | | |
| DMEM no glucose | Thermo-Fisher | 11966025 |
| Advanced DMEM/F-12 | Gibco | 12634010 |
| Neurobasal media | Thermo-Fisher | 21103049 |
| KnockOut DMEM | Thermo-Fisher | 10829018 |
| KnockOut Serum Replacement (KSR) | Gibco | 10828028 |
| Opti-MEM | Gibco | 31985062 |
| L-Glucose | Sigma-Aldrich | G8270 |
| L-Galactose | Sigma-Aldrich | G7134 |
| l-Glutamine | Sigma-Aldrich | G7513 |
| MEM nonessential amino acid solution | Sigma-Aldrich | M7145 |

| Reagent/resource | Reference or source | Identifier or catalog number |
|---|---|---|
| 2-Mercaptoethanol | Thermo-Fisher | 31350010 |
| Sodium Pyruvate | Thermo-Fisher | 11360070 |
| Bovine serum albumin solution (BSA) | Sigma-Aldrich | A8412 |
| Trypsin | Sigma-Aldrich | T4799 |
| Accutase | Sigma-Aldrich | SF006 |
| DPBS | Sigma-Aldrich | D8537 |
| Fetal bovine serum | Generated in-house | N/A |
| LIF | Generated in-house | N/A |
| N2 Supplement | Thermo-Fisher | 17502048 |
| B27 Supplement | Thermo-Fisher | 17504044 |
| Nicotinamide (NAM) | Sigma-Aldrich | N3376 |
| PD0325901 | Axon Medchem | 1408 |
| CHIR99021 | Sigma-Aldrich | SML1046 |
| G418 | InvivoGen | ant-gn-1 |
| Lipofectamine 2000 | Thermo-Fisher | 11668027 |
| 6-well plates | Sigma-Aldrich | CLS3516 |
| 10 cm dishes | Sigma-Aldrich | CLS430293 |
| Hoechst 33342 | Invitrogen | H3570 |
| DAPI | Sigma-Aldrich | D9542 |
| **Software** | | |
| LAS X | Leica Microsystems | |
| Fiji | ImageJ | |
| ImarisX64 9.5.1 | Oxford Instruments | |
| FCS Express v6 | BD Biosciences | |
| BD FACSDiva™ | BD Biosciences | |
| Metaboanalyst 2.0 | Chong et al, 2019 | |
| MaxQuant v 1.6.0.1, 1.6.15 | MaxQuant | |
| Perseus v 1.6.0.2, 1.6.14 | MaxQuant | |
| IGV_2.14.0 | Broad Institute | |
| **Other** | | |
| SH800S Cell sorter | Sony Biotechnology | |
| LSR Fortessa | BD bioscience | |
| SP8 confocal microscope | Leica | |

## Cell culture

ESC lines were maintained in complete ESC medium: DMEM no glucose (Gibco #11966025) supplemented with 10% FBS (Gibco), 4.5 g/L D-glucose (Sigma-Aldrich), 100 mM 2-mercaptoethanol (Sigma-Aldrich), 100 mM MEM nonessential amino acids, 2 mM L-glutamine, 1 mM sodium pyruvate (all from Gibco), and 1000

units/ml LIF (made in-house) on gelatinized tissue culture flasks (Corning). EMM is as described above, but without the addition of glucose and pyruvate, and with 4.5 g/L D-galactose. For 2i/LIF and KOSR/LIF culture, ESCs were maintained, as previously described in Martin Gonzalez et al (Martin Gonzalez et al, 2016).

Neural monolayer differentiation was performed as outlined by (Ying et al, 2003). In brief, cells were collected, washed in N2B27, and seeded at a density of $10^4/cm^2$, with daily media changes. Cells were cultured for 7 days before analysis of differentiation by RT-qPCR. To differentiate ESCs toward PrE, cells were cultured as previously described in (Anderson et al, 2017). In brief, ESCs were plated at $5 \times 10^4$ cells/cm² onto gelatinized plates in endoderm base medium, (RPMI 1640 medium with Glutamax (Gibco), supplemented with B27 minus insulin (Gibco A1895601)) for 24 h. Activin A (20 ng/ml) and CHIR99021 (3 μM) were added to base medium for subsequent culture, and the media was changed every other day. Cells were cultured for 5 days before analysis of differentiation by RT-qPCR. To differentiate ESCs toward trophoblast, ESCs were cultured as previously described in (Morgani et al, 2013). In brief, $10^4$ cells were plated per well of a six-well plate into trophoblast stem cell medium (70% mouse embryonic fibroblast-conditioned medium (R&D) and 30% TS cell medium [RPMI, GIBCO; glutamine and sodium pyruvate, GIBCO; 0.1 mM 2-mercaptoethanol, Sigma; 20% fetal calf serum). A total of 25 ng/ml Fgf4 (Peprotech) was added to the medium along with 1 mg/ml heparin sulfate (Sigma). Cells were cultured for 7 days before analysis of differentiation by RT-qPCR.

## Generation of Sirt1-KO and Sirt1-FKBP cell lines

Sirt1 CRISPR-KO cell lines were generated in mouse E14Ju ESCs from the 129/Ola background (Hamilton and Brickman, 2014) excising the catalytic domain of Sirt1 with guide RNAs that targeted Exon4 and Exon6 of Sirt1 (see Appendix Fig. S5a). Sirt1-FKBP ESC lines were generated by adding a Sirt1-GFP-FKBP construct (see Appendix Fig. S5f), downstream of a Tet-inducible promoter, into Sirt1-KO ESCs. All cell lines were checked regularly for appropriate antibiotic resistance, and in the case of Sirt1-FKBP, cells were checked by flow cytometry for GFP expression to ensure correct expression.

## Seahorse assays

ESCs were seeded into 96-well Seahorse plates precoated with gelatin at $40 \times 10^4$ cells per well. Oxygen Consumption Rate (OCR) and extracellular acidification rate (ECAR) were determined by the XF cell mito-stress test (#101706-100, Seahorse Biosciences) as described previously (Mahato et al, 2014), and basal measurements were taken from the third recorded time point. Values were normalized to total protein levels per well.

## Embryo culture

Oocytes were collected from the oviducts of prepubescent hormone-stimulated C57BL/6NRj females and fertilized in vitro according to CARD protocols, adapted by Infrafrontier (infra-frontier.eu/knowledgebase/protocols/cryopreservation-protocols). The resulting zygotes were cultured in KSOM medium (made in-house), which contained either D-glucose or D-galactose. These

embryos were then cultured for 5 days to reach the equivalent of an E4.5 in vivo embryo. Embryos were cultured in distinct microdrops for each condition, overlaid with embryo culture mineral oil (Sigma). Embryos were cultured at 37 °C, 5% $CO_2$ and 90% relative humidity. Embryos were then stained for NANOG (Epi marker) and CDX2 (TE marker) to analyze any differences between conditions. Animal work was carried in accordance with European legislation, authorized by the Danish National Animal Experiments Inspectorate (Dyreforsøgstilsynet, license no. 2023-15-0201-01513 and 2023-15-0202-00199) and performed according to national guidelines. Mice were maintained in a 12-h light/dark cycle in the designated facilities at the University of Copenhagen, Denmark.

## Morula injection and E6.5 contribution

E2.5 morula embryos were de-compacted in PB1 medium without calcium and magnesium for 20 min at room temperature, and 5 H2B-Tomato tagged ESCs, previously cultured in EMM, Serum/LIF or 2i/LIF, were introduced by microinjection. Resultant embryos were transferred to E0.5 pseudo-pregnant mothers. At E6.5, embryos were dissected from the decidua and contribution of H2B-Tomato ESCs was assessed. Embryos were stained for GATA6 (Visceral and Parietal endoderm marker) and KRT7 (Trophecto-derm marker).

## Alkaline phosphatase staining

ESCs were plated at clonal density and cultured for 9–10 days. Alkaline phosphatase staining was carried out with the diagnostic kit 86-R (Sigma) as per the manufacturer's instructions. Colonies were scored as 100% differentiated or undifferentiated or as mixed colonies containing both undifferentiated and differentiated cells. Colonies were imaged either using a Nikon AZ-100 microscope for 15-cm dishes or by scanning the six-well plates and were quantified using Fiji.

## Immunofluorescence

Cells were washed and fixed in 4% formaldehyde (Fisher Scientific, PI-28906), blocked, and permeabilized in 5% donkey serum and 0.3% Triton. Antibodies were incubated overnight in 1% BSA and 0.3% Triton in PBS and subsequently visualized with the appropriate secondary antibody (Alexa Fluor, Molecular Probes) and imaged using a Leica TCS SP8 confocal microscope and analyzed using Imarisx64 9.5.1 software.

## Western blot analysis and immunoprecipitation

Blotting was performed as previously described (Hamilton et al, 2013), except that primary antibodies were detected using fluorescently conjugated secondary antibodies (Alexa Fluor, Molecular Probes), visualized using a Chemidoc MP (Bio-Rad) and quantified using Fiji.

For immunoprecipitation, ESCs were cultured in Serum/LIF, EMM or EMM + NAM for 24 h, washed in ice-cold PBS, lysed in Young lysis buffer 1 (YB1: 50 mM HEPES KOH, pH 7.5, 140 mM NaCl, 1 mM EDTA, 10% glycerol, 0.5% NP-40 and 0.25% Triton X-100, supplemented with 1× cOmplete™ EDTA-free protease inhibitor tablets (Roche) and 10 mM Sodium Butyrate), then sonicated. Immunoprecipitation was performed against 10 μg

protein with 1 µg antibody in YB1. Antibody–protein complexes were collected with the appropriate magnetic bead and extensively washed in YB1 and eluted in 1× Laemmli buffer. Samples were then analyzed by western blot.

## Fluorescence-activated cell sorting (FACS) and flow cytometry

Cells were dissociated with Accutase (A6964, Sigma) and incubated with the appropriate antibody in 10% FCS:PBS for 30 min, washed extensively, and analyzed on an LSR Fortessa (BD Biosciences). Dead cells were excluded based on DAPI inclusion. For FACS followed by Seahorse assays, Hhex-Venus ESCs were stained for PECAM-1 as described previously (Canham et al, 2010), and sorted with a BD FACS Aria III cell sorter. Cells were collected in 10% FCS:PBS, then reconstituted in Serum/LIF, 2i/LIF, or KOSR/LIF and seeded in Seahorse plates, as described above. For flow cytometry, cells were treated as above and analyzed using a BD LSR Fortessa. Data was analyzed with FCS Express 6 software (BD Biosciences). Histograms shown in the manuscript are for 10,000 events, gated for DAPI-negative signal.

## Real-time qPCR

The total RNA was collected using either Trizol (Invitrogen) or the RNeasy Mini Kit (QIAGEN). Genomic DNA was eliminated by DNase treatment (QIAGEN), and 1 µg of total RNA was used for first-strand synthesis with SuperScript III reverse transcriptase according to the manufacturer's instructions. cDNA corresponding to 10 ng total RNA was used for real-time (RT)-qPCR analysis using the Roche LC480, and target amplification was detected with the Universal Probe Library system. Values were normalized to the geometric mean for *GAPDH*, *Pgk1* and *Sdha* expression.

## RNA-seqs

Total RNA was purified by standard methods and rRNA depleted using the Ribo-Zero kit (Illumina, as per the manufacturer's instructions). Libraries were prepared for Illumina sequencing using the NEBNext Ultra kit as per the manufacturer's instructions. For all conditions, three biological replicate samples were collected from independent experiments. RNA-seq libraries were prepared on-bead using the NEBNext Ultra kit as per the manufacturer's instructions and subsequently sequenced using a Next-Seq 500 Sequencer (Illumina).

## ATAC-seq

ATAC-seq was performed following methods previously described (Buenrostro et al, 2013). Adherent cells were treated with Accutase to obtain a single-cell suspension. Cells were counted and resuspended to obtain 50,000 cells per sample in ice-cold PBS. Cells were pelleted and resuspended in lysis buffer (10 mM Tris-HCl pH 7.4, 10 mM NaCl, 3 mM MgCl₂, 0.1% IGEPAL). Following a 10 min centrifugation at 4 °C, nucleic extracts were resuspended in transposition buffer for 30 min at 37 °C and purified using a QIAGEN MinElute PCR Purification kit following the manufacturer's instructions. Transposed DNA was eluted in a 10 mL volume and amplified by PCR with Nextera primers (Buenrostro

et al, 2013) to generate paired-indexed libraries. A maximum of 12 cycles of PCR was used to prevent saturation biases based on optimization experiments performed using RT-qPCR. Library quality control was carried out using the Bioanalyzer High-Sensitivity DNA analysis kit. Libraries were sequenced as paired-end 50 bp reads, sequenced using a Next-Seq 500 Sequencer (Illumina). For all conditions, two biological replicate samples were collected from independent experiments.

## Metabolome

Cells (500,000 per sample) were cultured in Serum/LIF or EMM for 3 h, 9 h, and 24 h, then spun down at 500 × *g* for 3 min, aspirated, and resuspended in 1 ml 0.9% NaCl. Samples were spun down and aspirated again, then stored at −80 °C until further processing. Metabolite extraction from the frozen pellets was then performed by adding 200 µL pre-chilled 50%MeOH with 0.0007 mg/ml D5-tryptophan, followed by sonication (5 °C for 2 rounds, 30 s each) and centrifugation (0 °C at 15,000 × *g* for 10 min). In total, 150 µl supernatant was collected and frozen at −80 °C, with 10 µL from each sample collected in one tube to make a Quality Control (QC). Pellets were reconstituted with 250 µL MilliQ H₂O, then sonicated and the protein concentration was measured using the Bicinchoninic acid (BCA) assay (Pierce).

LC-MS analysis was performed as described before by (Dall et al, 2018), with few modification of the A and B solvent composition, injection volume and MSMS analysis. Briefly, metabolite extracts, QC samples and blanks were defrosted on ice, vortexed and set in a pre-chilled vial. Samples were maintained at 4 °C throughout the analysis. Chromatographic separation was performed using UHPLC Dionex Ultimate 3000 (Thermo Scientific, Germany) with Luna Polar C18 column (1.6 µm, 2.1 × 100 mm, Phenomenex, USA) with EVO C18 guard column (sub-2 µm, 2.1 mm, Phenomenex, USA) kept at 40 °C. Solvent A and B were 0.1% formic acid in acetonitrile and 0.1% formic acid in LC-MS grade water, respectively. A flow rate of 0.3 mL/min was applied with a gradient elution profile: 95% B 0–1 min, 95%–5% B 1.0–10.0 min, 5% B 10.0–12.0 min, 5–95% B 12.0–12.5 min, 95% B 12.5-14-5 min. LC was coupled with QToF Impact II mass spectrometer (Bruker Daltonics, Germany). Samples were analyzed in positive and negative mode. In total, 10 µL of the extract was injected in the positive mode and 20 µL in the negative mode. MS spectra were acquired in the mass range 50–1000 mass to charge ratio (*m/z*) at 2.00 Hz spectra rate using the source settings for positive mode: absolute threshold 50 cts per 1000 sum, End Plate Offset 500 V, Capillary 4500 V, Nebulizer 2.0 Bar, Dry Gas 10.0 l/min, Dry Temperature 220 °C; Transfer: Funnel 1RF 150.0 Vpp, Funnel 2FR 200.0 Vpp, isCID Eergy 0.0 eV, Hexapole RF 50.0 Vpp; Quadrupole: Ion Enegry 4.0 eV, Low Mass 100.0 *m/z*; Collision Cell: Collision Energy 7.0 eV, Transfer Time 65.0 µs, Collision RF 650.0 Vpp, Pre Pulse Storage 5.0 µs. In negative mode Capillary voltage was set to 3000 and other parameters were identical as described above for both modes.

MSMS analysis of the QC samples for metabolite identification purposes was performed at the same LC-MS settings as the MS scans and collision energy set to 25 for both negative and positive modes. Raw data are available through MetaboLights platform: www.ebi.ac.uk/metabolights/MTBLS2691 (identification number of the study MTBLS2691) (Haug et al, 2020). Raw data from the positive

and negative mode were analyzed separately using MetaboScape 4.0 (Vasilopoulou et al, 2020) (Bruker Daltonics, Germany). Mass calibration was based on sodium formate clusters and lock-mass calibration with hexakis(2,2-difluoroethoxy)phosphazene (Apollo Scientific Ltd, UK). Feature detection was performed using an intensity threshold of 1000 counts in the positive mode and 500 in the negative. Metabolites annotation of detected molecular features with assigned MS/MS spectra was performed using HMDB Metabolite Library and MetaboBASE Personal MSMS library (Bruker Daltonics, Germany). Identified compounds were inspected manually for peak shape, retention time and structure. Final data were normalized by the intensity of internal standard within each sample and multiplied by its average value in the samples. Molecular feature was retained in the final data, if its intensity was over double compared to the blanks injections and its reproducibility was within 30% of CV in QC samples. Data were analyzed using Metaboanalyst 2.0 software (Chong et al, 2019).

## Fatty acid oxidation detection

FAO detection was performed as previously described, adapted to use in mouse ESCs (Dall et al, 2018). Briefly, mouse ESCs were plated at a density of 100,000 cells/well in a gelatin-coated 96-well plate. $^{14}$C-labeled palmitate/BSA complexes were then prepared as described previously (Dall et al, 2019), then added to Serum/LIF and EMM media. Overall, 30 μl of 1M NaOH/well was added to a glass-fiber filter 96-well plate (catalog no. 6005199, PerkinElmer Life Sciences), and clamped on top of the culture plate, separated by a custom-made perforated rubber insert. The back of the filter plate was sealed with a back cover seal (catalog no. 6005199, PerkinElmer Life Sciences), and the plates were wrapped in parafilm. The mESCs were incubated at 37 °C for 4 h, and the filter plate was recovered to detect $^{14}$CO$_2$ release. The filter plate was dried at 50 °C for 1 h. Counts from wells with cells stimulated with media without [$^{14}$C] palmitate were subtracted as background. Wells with $^{14}$C-media but without cells were included as an internal control and showed the background level of counts/min on the filter (data not shown).

## MS of histone PTMs

Cells were seeded in 15-cm dishes ($5 \times 10^6$ cells per dish, one dish per cell line) and collected 2 d later by trypsinization and washing in PBS. Cell pellets ($1 \times 10^7$ cells) were snap-frozen and shipped on dry ice to EpiQMAx GmbH. Sample preparation and MS analysis were performed according to the EpiQMAx GmbH protocols (https://www.epiqmax.com/service#solutions).

## Proteome and acetylome

### Sample preparation
For proteome analysis, four mESC lines (E14-early passage (ep), E14-late passages (lp), E14-ZscancE, E14-FUCCI) served as biological replicates and were each cultured in either Serum/LIF (SL), 2iLIF (N2B27, CHIR99021, PD03, LIF), or KOSR/LIF (Gibco) for more than three passages. Cell pellets were collected and frozen at −20 °C. For lysis, pellets were thawed on ice and resuspended in 6 M Guanidinium Hydrochloride (GndCl) with 5 mM TCEP, 10 mM CAA, 100 mM Tris pH 8.5 and heated to 99 °C for 10 min. Lysates were sonicated and then digested with LysC

(Wako), (enzyme:protein ratio of 1:100 (w/w)) for 2 h at room temperature (RT), followed by dilution with 25 mM Tris pH 8.5 to 2 M GndCl and further digested overnight with trypsin (Sigma-Aldrich) at 1:100 (w/w) at 37 °C. Peptides were basified to 40 mM ammonium hydroxide then purified by StageTip (C18 material) high pH fractionation. For this, StageTips were first activated in 100% methanol, then equilibrated in 80% acetonitrile in 50 mM ammonium hydroxide, and finally washed twice in 50 mM ammonium hydroxide. Samples were loaded on the equilibrated StageTips and washed twice with 50 mM ammonium hydroxide. StageTip fractionation elution was performed with 40 μL of 50 mM ammonium hydroxide containing increasing amounts of acetonitrile (5, 10, 15, 20, 27, 35%). Eluted sample fractions were dried to completion in a SpeedVac at 60 °C, dissolved in 12 μL 0.1% formic acid, and stored at −20 °C until MS analysis.

For lysine acetylation analysis, mESCs were grown in conventional Serum/LIF media and switched to media with D-galactose (Sigma) replacing D-glucose and pyruvate (labeled Gal), with or without 20 mM NAM (labeled NAM) and collected after 24 h and compared to cells with a media change of conventional Serum/LIF (labeled Glu). Cells were washed three times in ice-cold PBS, before being harvested and resuspended with 10 volumes of 6 M Guanidinium Hydrochloride (GndCl in 50 mM Tris pH 8.5) and snap-frozen in liquid nitrogen. Pellets were then thawed, reduced and alkylated in 5 mM TCEP and CAA at room temperature (RT), before sonication and measurement of protein content by Bradford (QIAGEN). Proteins were then digested with LysC (1:200) for 3 h at RT, then dilution with 3 volumes of ammonium bicarbonate (ABC), and incubated with trypsin (1:200) overnight at RT. Peptides were then acidified to 0.5% TFA and loaded onto a SepPak C18 Classic Cartridge (VWR), eluted with 30% ACN in 0.1% TFA, frozen, and lyophilized for 5 days. To enrich for acetylated peptides, samples were incubated with PTMScan Acetyl-Lysine (Ac-K) beads (CST) for 2 h at 4 °C, following the commercial protocol for buffer, wash, and elution conditions. Peptides were then purified by StageTip (C18 material) and fractionated at high pH as described above. Specifically, samples were eluted with 75 μl of 2, 4, 7,10, 15, or 25% ACN in 50 mM ammonium hydroxide. Flow-through during sample loading was collected, acidified and reloaded onto a new StageTip that had been activated with 30% ACN in 0.1% formic acid, washed two times with 0.1% formic acid. This fraction (F0) was then eluted with 75 μL of 25% ACN in 0.1% formic acid, then dried and reconstituted the same as the other fractions.

## Mass spectrometry

MS samples were analyzed on an EASY-nLC 1200 system (Thermo) coupled to either a Q Exactive HF-X Hybrid Quadrupole-Orbitrap mass spectrometer (Thermo) for the proteome analysis or an Orbitrap Exploris 480 mass spectrometer (Thermo) for acetylome analysis. Separation of peptides was performed using 15 cm columns (75 μm internal diameter) packed in-house with ReproSil-Pur 120 C18-AQ 1.9-μm beads (Dr. Maisch). Elution of peptides from the column was achieved using a gradient ranging from buffer A (0.1% formic acid) to buffer B (80% acetonitrile in 0.1% formic acid), at a flow rate of 250 nl/min. Gradient length was 80 min per sample (proteome) and 70 min per sample (acetylome), including ramp-up and washout. The column was heated to 40 °C using a column oven, and ionization was achieved using a NanoSpray Flex ion source (Thermo). Spray voltage set at 2 kV, ion transfer tube temperature to 275 °C, and RF funnel level to

40%. Measurements were performed with a full scan range of 300–1750 *m/z*, MS1 resolution of 60,000 (HF-X) or 120,000 (Exploris), MS1 AGC target of 3,000,000 (HF-X) or normalized AGC target of 200% (Exploris) and MS1 maximum injection time of 60 ms (HF-X) or auto (Exploris). Precursors with charges 2–6 were selected for fragmentation using an isolation width of 1.3 *m/z* and fragmented using higher-energy collision disassociation (HCD) with a normalized collision energy of 25%. Precursors were excluded from re-sequencing by setting a dynamic exclusion or 60 s. MS2 AGC target was set to 200,000 and minimum MS2 AGC target to 20,000 (HF-X) or normalized AGC target of 200% (Exploris). For proteome runs, MS2 maximum injection time was 60 ms, MS2 resolution was 30,000, and loop count was 12. For acetylome runs, a MS2 maximum injection time of auto, MS2 resolution of 60,000, and loop count of 7.

## CUT&Tag

CUT&Tag was performed as previously described (Kaya-Okur et al, 2019, 2020) with slight alterations. All SIRT1-FKBP samples were isolated by FACS based on GFP levels (except Sirt1-dTag samples), and $2 \times 10^5$ cells per sample replicate were used for downstream processing. Primary antibody incubation was done at 4 °C overnight, Tn5 was purchased from EMBL Heidelberg, and DNA precipitation was performed over the weekend at −80 °C. DNA was amplified with 12 PCR cycles and the samples were sequenced paired end on an Illumina Next-Seq 500.

## Model development for SOX2 binding to chromatin

The objective of the model was to investigate if it is possible to obtain an increase in *specific binding* of pluripotency transcription factors, TF, (e.g., Sox2) as a result of hypothesized, SIRT1-mediated increase in binding affinity of histones, $K_h$, and pluripotency transcription factors, $K_s$.

The specificity of binding, which we denote as specificity, is given by the probability of the specific site to be bound to the pluripotency enhancers, here referred to as specific sites, $N_s$

$$Specificity = [T:N_s]/N_s = \frac{\left(T_f/K_s\right)^2}{1 + \left(T_f/K_s\right)^2 + H_f/K_h} \quad (1)$$

We assume that TFs bind to specific sites $N_s$ cooperatively, with hill coefficient 2. In addition TF can bind to unspecific sites, $N_u$, albeit with different dissociation constants, $K_s$ and $K_u$, and the specific binding is tenfold stronger than unspecific $K_s = 0.1K_u$.

We also assumed that the number of unspecific binding sites is in excess of the specific $N_u >> N_s$, and that the total number of histones, H, is in excess over the total number of TFs, H >> T.

The expression for specificity has two unknowns, $T_f$ and $H_f$, which can be found from mass conservation equations. Given that TFs and histones can bind to both $N_s$ and $N_u$, we can write the mass conservation equations for H, T, $N_s$ and $N_u$.

Expressing the complexes through free concentrations and binding affinities, and using the assumptions $N_u >> N_s$, and H >> TF, we obtain four equations with four unknown free concentrations for histones, $H_f$, TFs, $T_f$; specific sites $N_{sf}$; and unspecific sites, $N_{uf}$. These equations were solved with the help of Python library SymPy, and the resulting expressions for $T_f$ and $H_f$, were used for calculating specificity in Eq. (1). The estimates of

parameter values and related references are given in the following table:

| Parameter | Value | Description | Reference |
|---|---|---|---|
| H | 10 µM | Total histone concentration | Hihara et al., 2012 |
| T | 0.1 µM | Total TF concentration | Strebinger et al, 2019 |
| N_s | 0.01 µM | Total specific site concentration | Boyer et al, 2005 |
| N_u | 10 µM | Total unspecific site concentration | Boyer et al, 2005 |
| K_s | 1–100 nM | TF dissociation constant to specific sites | Mistri et al, 2018 |
| K_u | 10*K_s | TF dissociation constant to unspecific sites | Kolesov et al, 2007 |
| K_h | 1–100 nM | Histone dissociation constant | Caterino et al, 2011; White et al, 2016 |

## Statistical analysis

Quantitative data are presented as mean ± s.d. unless otherwise indicated. Statistical analyses were performed using Prism software (GraphPad). Details for statistical analyses, including replicate numbers, are included in figure legends.

### Data analysis for RNA- and ATAC-seq

For RNA-seq, sequencing reads (60 bases) were aligned using the STAR package. Allocation of reads at introns (and exons) were examined using Table Browser (UCSC) to define the corresponding genomic intervals. Reads per gene per class were counted using HTSeq with the categorized alignment files as input. Genes were considered significantly regulated if they exhibited $Log_2FC > 1$, and $P_{adj} < 0.01$.

For ATAC-seq, regions were considered significantly regulated if they exhibited $Log_2FC > 1$, and $P_{adj} < 0.01$. Metaprofiles were generated from bigWig files using deepTools software (Ramírez et al, 2014). Enhancer gene association was performed using GREAT (http://great.stanford.edu/).

Data processing and analysis were performed using Computerome, the National Life Science Supercomputer at DTU (www.computerome.dk), and the Bioconductor package, DESeq2 (Love et al, 2014).

The IMAGE pipeline was performed to identify active motifs from our ATAC-seq and RNA-seq data. The pipeline was obtained from (Grud et al, 2018), and is also available on GitHub (https://github.com/JesperGrud/IMAGE). The IMAGE pipeline required as input: our ATAC-seq data normalized by library size, our RNA-seq raw count matrix, and the primary assembly genome of mm10 from GENCODE in fasta format. Pipeline results were processed using custom scripts derived from the IMAGE pipeline GitHub (https://github.com/JesperGrud/IMAGE).

### Data analysis for proteome and acetylome

All MS RAW data were analyzed using the freely available MaxQuant software (Cox & Mann, 2008), v. 1.6.0.1 (proteome) and v. 1.6.15 (acetylome) using the Andromeda search engine, and subsequent analysis was performed using Perseus v. 1.6.0.2

(proteome) or v.1.6.14 (acetylome). For generation of theoretical spectral libraries, the *Mus musculus* FASTA database was downloaded from UniProt on the 25 July, 2017 (proteome) or the 21 July, 2020 (acetylome). In silico digestion of proteins to generate theoretical peptides was performed with trypsin, allowing up to three missed cleavages. Allowed variable modifications were oxidation of methionine (default), protein N-terminal acetylation (default) for all samples. Additional variable modifications included Acetyl (K) and Phospho (STY). The second peptide search was enabled. Matching between runs was enabled with an alignment window of 20 min and a match time window of 0.7 min. For proteome, Label-free quantification (LFQ) was enabled based on (Cox et al, 2014) and maximum variable modifications per peptide were reduced to 3. Stringent MaxQuant 1% FDR data filtering at the PSM- and protein levels was applied (default). For acetylome analysis, peptide intensities were median normalized in Perseus. Proteins or acetylated peptides only identified by one site and potential contaminants were removed and then data was filtered for proteins or sites identified in at least 75% of the replicates in at least one condition. Conditions were compared using a two-way ANOVA (proteome) or two sample Student's *t* tests (acetylome) with 5% FDR. Cytoscape software was used with the String App and Omics Visualizer to create protein networks with functional enrichment of GO Terms and KEGG pathways.

### Data analysis for CUT&Tag

Reads were aligned to the mm10 primary assembly using Bowtie2 (Langmead and Salzberg, 2012). Reads were quality filtered using Samtools (Danecek et al, 2021), and duplicates marked and removed using Picard (Picard Toolkit." 2019. Broad Institute, GitHub Repository. https://broadinstitute.github.io/picard/). Fragment bedgraphs were produced using DeepTools bamCoverage (Ramírez et al, 2016). Peak calling was performed using SEACR (Meers et al, 2019). Peak calling by Sparse Enrichment Analysis for CUT&RUN chromatin profiling. SEACR relaxed was used for transcription factors, and SEACR stringent was used for histone marks, using normalized mode with condition-matched IgG samples as controls. Peaks called in 2/3 replicates for Tead1 and Klf4, and in 4/6 replicates over two experiments for Sox2 were used. Active enhancers were defined as regions with called peaks for both H3K27ac and H3K4me1 in all replicates. Read fragments at ± 200 bp regions centered on enhancer summits were counted using Bedtools map (Quinlan and Hall, 2010).

## Data availability

The ATAC-seq and RNA-seq data have been deposited in the Gene Expression Omnibus, under the accession number GSE173543. The mass spectrometry metabolome data have been deposited with the Metabolites repository with the dataset identifier MTBLS2691. The mass spectrometry proteome and acetylome data have been deposited with the ProteomeXchange Consortium via the PRIDE partner repository with the dataset identifier PXD025468.

The source data of this paper are collected in the following database record: biostudies:S-SCDT-10_1038-S44318-025-00417-0.

## Peer review information

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

## Acknowledgements

The authors thank the reNEW Genomics Platform (M Michaut, H Wollmann), reNEW Flow Cytometry Platform (G dela Cruz, P van Dieken), Stem Cell

Culture Platform (EF Rebollo), reNEW Imaging Platform (J Bulkescher), the Core Facility for Transgenic Mice (J Martin Gonzalez), the Mass Spectrometry Platform (M Rykær), for technical expertise, support and the use of instruments, the Brickman laboratory members for critical discussions, Madeleine Linneberg-Agerholm, Henrik Semb and Jan Zylicz, for comments. Raw sequencing data were converted from bcl to fastq using the DeiC National Life Science Supercomputer at DTU (www.computerome.dk). The SOX2-FKBP cell line was a kind gift from the Elzo de Wit lab. Work in the Brickman lab was supported by Lundbeck Foundation (R198-2015-412, R370-2021-617 and R400-2022-769), Independent Research Fund Denmark (DFF-8020-00100B, DFF-0134-00022B, and DFF-2034-00025B), European Union (ERC, SENCE, 101097979) and the Danish National Research Foundation (DNRF116). ARR was supported by Lundbeck Foundation PhD studentships (R208-2015-2872). The Novo Nordisk Foundation Center for Stem Cell Medicine (reNEW) is supported by a Novo Nordisk Foundation grant number NNF21CC0073729, and previously NNF17CC0027852. Work in the Trusina lab is supported by the Danish National Research Foundation (DNRF116). The Novo Nordisk Foundation Center for Basic Metabolic Research (CBMR) is an independent Research Center at the University of Copenhagen, which is partially funded by an unrestricted donation from the Novo Nordisk Foundation (NNF18CC0034900). The Novo Nordisk Foundation Center for Protein Research (CPR) is funded in part by a donation from the Novo Nordisk Foundation (NNF14CC0001 and NNF24SA0098829).

## Author contributions

**Robert A Bone**: Conceptualization; Data curation; Formal analysis; Validation; Visualization; Methodology; Writing—original draft; Writing—review and editing. **Molly P Lowndes**: Data curation; Formal analysis. **Silvia Rainieri**: Data curation; Formal analysis. **Alba R Riveiro**: Data curation; Formal analysis. **Sarah L Lundregan**: Formal analysis. **Morten Dall**: Formal analysis. **Karolina Sulek**: Data curation; Formal analysis. **Jose A H Romero**: Formal analysis. **Luna Malzard**: Data curation. **Sandra Koigi**: Formal analysis. **Indra J Heckenbach**: Formal analysis. **Victor Solis-Mezarino**: Data curation; Formal analysis. **Moritz Völker-Albert**: Formal analysis. **Catherine G Vasilopoulou**: Software. **Florian Meier**: Software. **Ala Trusina**: Supervision. **Matthias Mann**: Software. **Michael L Nielsen**: Supervision. **Jonas T Treebak**: Supervision. **Joshua M Brickman**: Conceptualization; Resources; Supervision; Funding acquisition; Investigation; Writing—original draft; Writing—review and editing.

Source data underlying figure panels in this paper may have individual authorship assigned. Where available, figure panel/source data authorship is listed in the following database record: biostudies:S-SCDT-10_1038-S44318-025-00417-0.

## Disclosure and competing interests statement

The authors RAB and JMB have filed a patent on the use of EMM for stem cell culture. The remaining authors declare no competing interests.

# Expanded View Figures

**Figure EV1.   Differentiation potential characteristics of EMESCs and effects of galactose on pre-implantation development.**

(A) Western blot analysis of SL-, EMM- and 2i/LIF-cultured mESCs. $n = 6$ biologically independent samples. Data are mean $+$ s.d., unpaired two-tailed $t$ test. $P$ values: Active B-CATENIN SL vs EMM n.s.$=$0.8464, EMM vs 2i/LIF **$P = 0.0040$; pYAP1 SL vs EMM *$P = 0.0124$, EMM vs 2i/LIF n.s.$=$0.0559; pERK SL vs EMM ****$P < 0.0001$, EMM vs 2i/LIF ****$P < 0.0001$; pAKT SL vs EMM ****$P < 0.0001$, EMM vs 2i/LIF *$P = 0.0148$; pSTAT3 SL vs EMM **$P = 0.0069$, EMM vs 2i/LIF n.s. $P = 0.7369$. (B) RT-qPCR analysis of pluripotency genes *Oct4*, *Nanog* and *Tfcp2l1*; PrE genes *Gata4*, *Sox7* and *cMyc*; and TE genes *Gata3* and *Eomes*. $n = 3$ biologically independent samples. Data are mean $+$ s.d., unpaired two-tailed $t$ test. $P$ values: *Oct4* n.s.$=$0.4303; *Nanog* n.s.$=$ 0.0729; *Tcfp2l1* n.s.$=$ 0.1515; *cMyc* **$P = 0.0053$; *Gata4* **$P = 0.0097$; *Sox7* *$P = 0.0159$; *Eomes* *$P = 0.0297$; *Gata3* *$P = 0.0380$. (C) RT-qPCR analysis of pluripotency genes *Nanog* and *Oct4*, and neural genes *Nestin* and *Zfp521* at d0 and d7 of neural differentiation. ESCs and EMESCs were cultured in Serum/LIF or EMM respectively for 2 passages prior to d0. Data are mean $+$ s.d., $n = 3$ biologically independent samples, unpaired two-tailed $t$ test. $P$ values: *Nestin* n.s.$=$0.2267; *Zfp521* n.s.$=$ 0.1027. (D) RT-qPCR analysis of pluripotency gene *Oct4* and PrE genes *Gata4*, *Gata6* and *Pdgfra* at d0, d4 and d7 of PrE differentiation. ESCs and EMESCs were cultured in Serum/LIF or EMM respectively for 2 passages prior to d0. Data are mean $+$ s.d., $n = 3$ biologically independent samples, unpaired two-tailed $t$ test. $P$ values: *Gata6* **$P = 0.0084$; *Pdgfra* **$P = 0.0050$. (E) RT-qPCR analysis of pluripotency genes *Nanog* and *Oct4*, and TSC genes *Gata3* and *Elf5* at d0 and d7 of TSC differentiation. ESCs and EMESCs were cultured in Serum/LIF or EMM respectively for 2 passages prior to d0. Data are mean $+$ s.d., $n = 3$ biologically independent samples, unpaired two-tailed $t$ test. *Gata3* **$P = 0.0011$; *Elf5* *$P = 0.0389$. (F) Brightfield images of mESCs after 10 passages cultured in Serum/LIF, EMM, and 2i/LIF. Scale bar $= 100$ µm. (G–I) Comparitive expression of apoptosis (G), embryonic (H) and extra-embryonic (I) genes from transcriptomic data in mESCs cultured for 10 and 20 passages in Serum/LIF, EMM, and 2i/LIF. $n = 3$ biologically independent samples. Data are mean $+$ s.d., unpaired two-tailed $t$ test. $P$ values: *Bad* SL vs 2i/LIF P20 **$P = 0.0245$, EMM vs 2i/LIF P20 ****$P < 0.0001$; *Bak1* EMM vs 2i/LIF P10 **$P = 0.0012$, SL vs EMM P20 **$P = 0.0012$, SL vs 2i/LIF P20 **$P = 0.0039$, EMM vs 2i/LIF P20 ****$P < 0.0001$; *Bid* SL vs 2i/LIF P10 ****$P < 0.0001$, EMM vs 2i/LIF P10 ****$P < 0.0001$, SL vs EMM P20 ****$P < 0.0001$, SL vs 2i/LIF P20 ****$P < 0.0001$; *Caspase3* EMM vs 2i/LIF P10 *$P = 0.03913$, SL vs 2i/LIF P20 ****$P< 0.0001$, EMM vs 2i/LIF P20 ****$P < 0.0001$; *Caspase9* SL vs 2i/LIF P10 ****$P < 0.0001$, EMM vs 2i/LIF P10 ****$P < 0.0001$, SL vs 2i/LIF P20 ****$P < 0.0001$, EMM vs 2i/LIF P20 ****$P < 0.0001$; *Tnfrsfla* SL vs 2i/LIF P10 ***$P = 0.0001$, EMM vs 2i/LIF P10 ****$P < 0.0001$, SL vs 2i/LIF P20 ****$P < 0.0001$, EMM vs 2i/LIF P20 ****$P < 0.0001$; *Nanog* SL vs 2i/LIF P10 *$P = 0.0180$, EMM vs 2i/LIF P10 **$P = 0.0022$, SL vs EMM P20 ***$P = 0.0007$, SL vs 2i/LIF P20 **$P = 0.0086$, EMM vs 2i/LIF P20 ****$P < 0.0001$; *Nr0b1* SL vs EMM P10 ****$P < 0.0001$, EMM vs 2i/LIF P10 ****$P < 0.0001$, SL vs 2i/LIF P20 ****$P < 0.0001$, EMM vs 2i/LIF P20 ****$P < 0.0001$; *Pou5f1* SL vs EMM P10 ***$P = 0.0009$, EMM vs 2i/LIF P10 ***$P = 0.0002$; *Sall4*, SL vs EMM P10 ****$P < 0.0001$, SL vs 2i/LIF P10 ***$P = 0.0007$, EMM vs 2i/LIF P10 ****$P < 0.0001$, SL vs EMM P20 **$P = 0.0050$, EMM vs 2i/LIF P20 ****$P < 0.0001$; *Sirt1* SL vs EMM P10 ****$P < 0.0001$, EMM vs 2i/LIF P10 ****$P < 0.0001$, SL vs EMM P20 ****$P < 0.0001$, EMM vs 2i/LIF P20 ****$P < 0.0001$; *Sox2* SL vs EMM P10 ****$P < 0.0001$, EMM vs 2i/LIF P10 ****$P < 0.0001$, SL vs EMM P20 ****$P = 0.0001$, EMM vs 2i/LIF P20 ****$P = 0.0001$; *Col4a1*, SL vs EMM P10 ****$P < 0.0001$, SL vs 2i/LIF P10 ****$P < 0.0001$, EMM vs 2i/LIF P20 **$P = 0.0013$, EMM vs 2i/LIF P20 ****$P < 0.0001$; *Col4a2*, SL vs EMM P10 **$P = 0.0013$, SL vs 2i/LIF P10 ***$P = 0.0006$, EMM vs 2i/LIF P10 ****$P < 0.0001$, SL vs 2i/LIF P20 **$P = 0.0037$, EMM vs 2i/LIF P20 **$P = 0.0024$; *Dusp4*, SL vs 2i/LIF P10 ****$P < 0.0001$, EMM vs 2i/LIF P10 ****$P < 0.0001$, SL vs EMM P20 ****$P < 0.0001$, SL vs 2i/LIF P20 ****$P < 0.0001$, EMM vs 2i/LIF P20 ****$P < 0.0001$; *Gata6*, SL vs 2i/LIF P20 ****$P = 0.0081$, EMM vs 2i/LIF P20 ****$P = 0.0037$; *Lrp2*, SL vs EMM P10 *$P = 0.0131$, SL vs 2i/LIF P10 ****$P < 0.0001$, EMM vs 2i/LIF P10 *$P = 0.0113$, SL vs EMM P20 ****$P < 0.0001$, EMM vs 2i/LIF P20 ****$P < 0.0001$; *Sox7*, SL vs EMM P10 **$P = 0.0015$, EMM vs 2i/LIF P10 ***$P = 0.0003$, SL vs 2i/LIF P20 **$P = 0.0099$, EMM vs 2i/LIF P20 ***$P = 0.0006$. (J) Representative image of mouse litter, star indicates mice with contribution (left); Table depicting contribution effeciency of mESCs (two different cell lines) cultured in SL or EMM in generating chimeric mice from injected blastocysts (right).

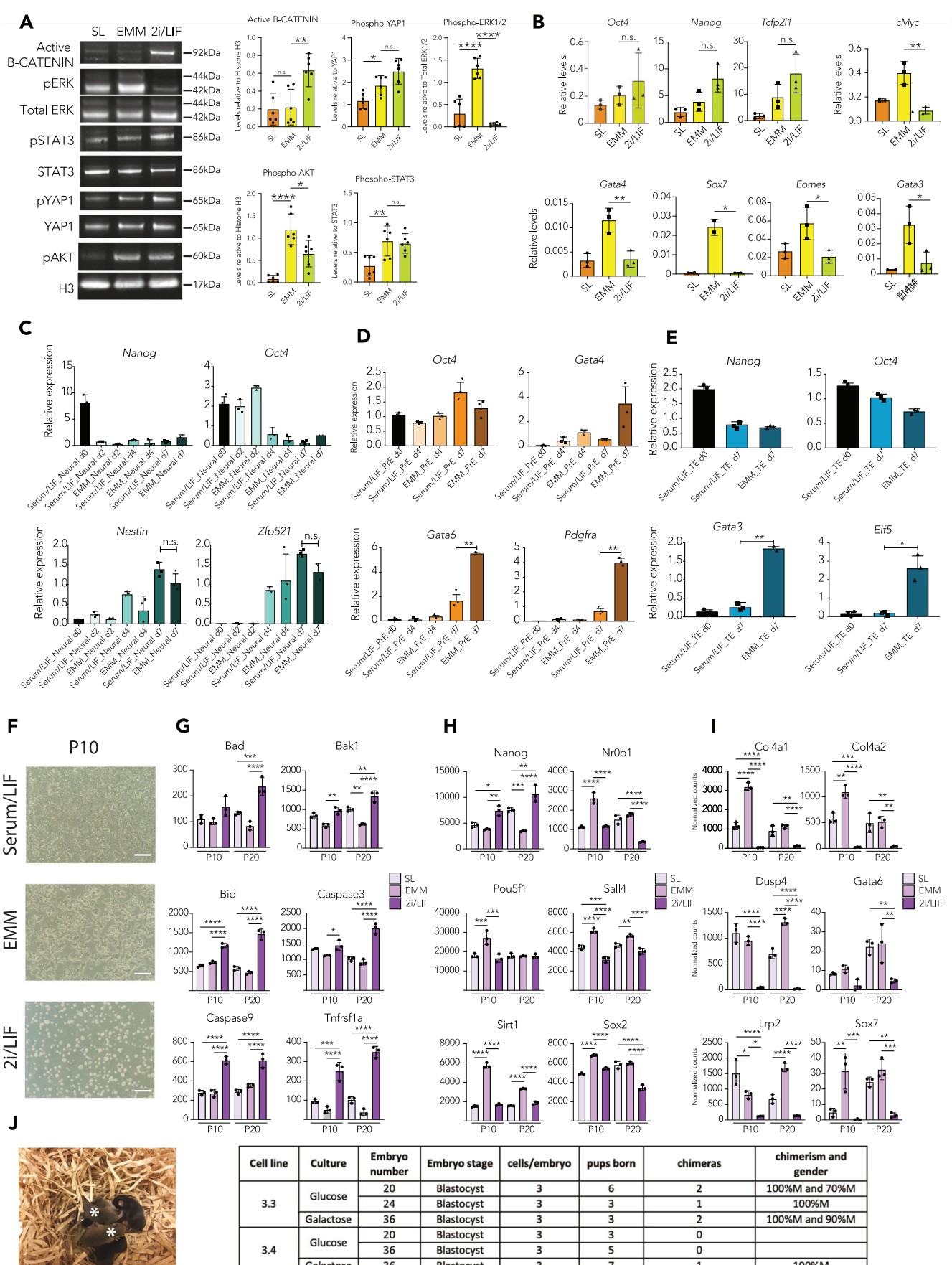

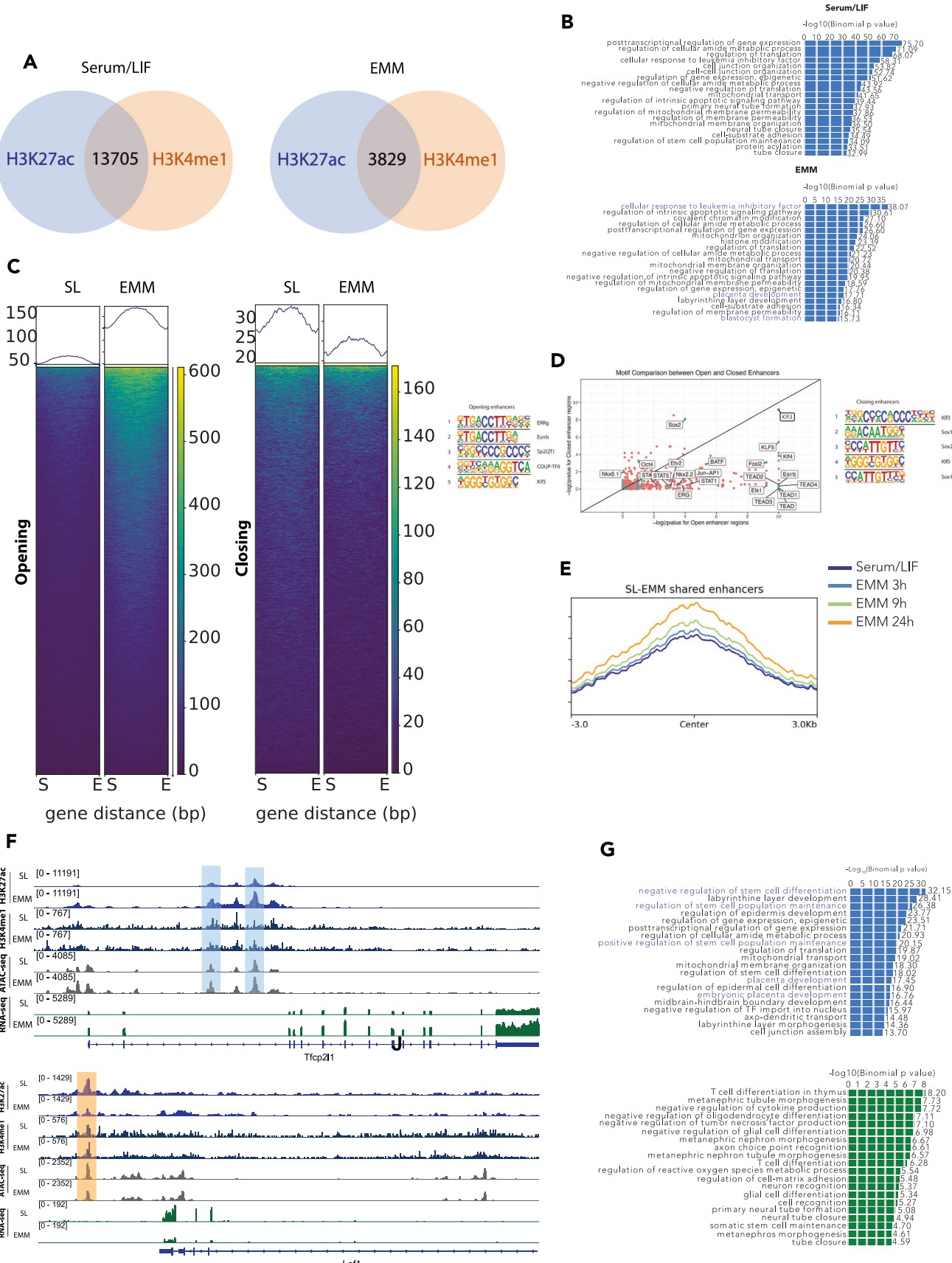

**Figure EV2. Further characterization of the enhancer landscape of EMM-cultured ESCs.**

(A) Overlap of H3K27ac and H3K4me1 CUT&Tag peaks, defining enhancers in Serum/LIF and EMM. (B) GO analysis for Biological Processes of genes associated with enhancers defined by overlapping peaks for H3K27ac and H3K4me1 in ESCs cultured in Serum/LIF (top) and EMM (bottom). Discussed GO terms highlighted in blue. (C) Heatmaps and profiles depicting H3K27ac intensity in both Serum/LIF and EMM conditions at opening and closing enhancers. (D) Motif enrichment analysis of Opening versus Closing enhancers. (E) ATAC-seq time course metaprofiles of SL-EMM shared enhancers. (F) Genome browser tracks (IGV 2.14.0 software) of CUT&Tag for H3K27ac and H3K4me1, ATAC-seq and RNA-seq at the Tfcp2l1 locus (top) and Lef1 locus (bottom) at 0 h and 24 h of EMM treatment. Blue boxes depict opening enhancers, orange box depicts a closing enhancer. (G) GO analysis for Biological Processes of genes associated with ATAC-upregulated (top) and -downregulated (bottom) TE regions after 24 h EMM culture (noted processes in blue font).

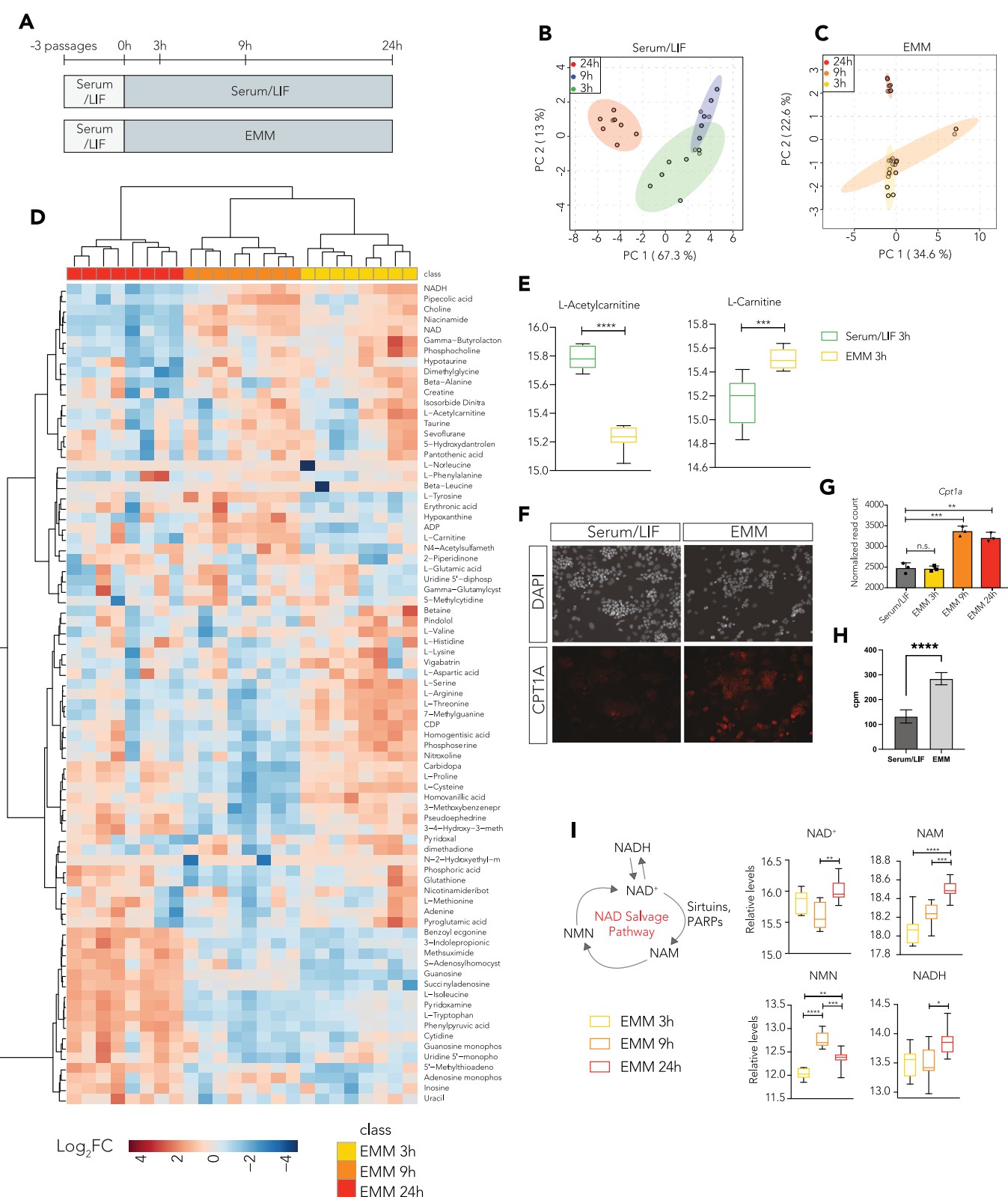

**Figure EV3.** **Downstream responses to EMM metabolic phenotypes and the enzymes propagating them.**

(A) Experimental outline for Serum/LIF and EMM time course used for metabolomic analysis. (B) PC1 vs 2 for Serum/LIF-cultured ESCs at 3 h, 9 h and 24 h in culture. (C) PC1 vs 2 for EMM-cultured ESCs at 3 h, 9 h and 24 h in culture. (D) Heatmap of enriched metabolites ($Log_2FC < 1$, $P_{adj} < 0.05$) in Serum/LIF-cultured ESCs versus all time points during EMM culture, $n = 8$ technical replicates per sample. (E) Levels of L-Acetylcarnitine (left) and L-Carnitine (right) in Serum/LIF and EMM-cultured ESCs after 3 h culture. P values: L-Acetylcarnitine ****$P < 0.0001$; L-Carnitine ***$P = 0.0004$, unpaired two-tailed $t$ test, $n = 8$ technical replicates per sample. (F) Immunostaining for CPT1A in Serum/LIF and EMM-cultured ESCs after 24 h. Data are representative of 3 biological replicates. Scale bar = 40μm. (G) Normalized counts from RNA-seq for *Cpt1a* in Serum/LIF and EMM-cultured ESCs after 3 h, 9 h and 24 h in the indicated condition. P values: n.s.=0.846 ***$P = 0.0008$, **$P = 0.0015$, unpaired two-tailed $t$ test, $n = 3$ biologically independent samples. (H) Fatty acid oxidation assay, measuring metabolism of $C^{14}$-labeled palmitic acid in SL- and EMM-cultured ESCs after 24 h, representative of 3 independent experiments. Data was normalized to protein levels. ****$P < 0.0001$, unpaired two-tailed $t$ test. (I) Schematic for $NAD^+$ salvage pathway with individual plots for the depicted metabolites after 3 h, 9 h, 24 h in EMM culture. P values: NAD + 9 h vs 24 h **$P = 0.001$; NAM 3 h vs 24 h ****$P < 0.0001$, 9 h vs 24 h ***$P = 0.0003$; NMN 3 h vs 9 h ****$P < 0.001$, 3 h vs 24 h **$P = 0.0012$, 9 h vs 24 h ***$P = 0.0009$; NADH 9 h vs 24 h *$P = 0.0145$; unpaired two-tailed $t$ test, $n = 8$ technical replicates per sample.

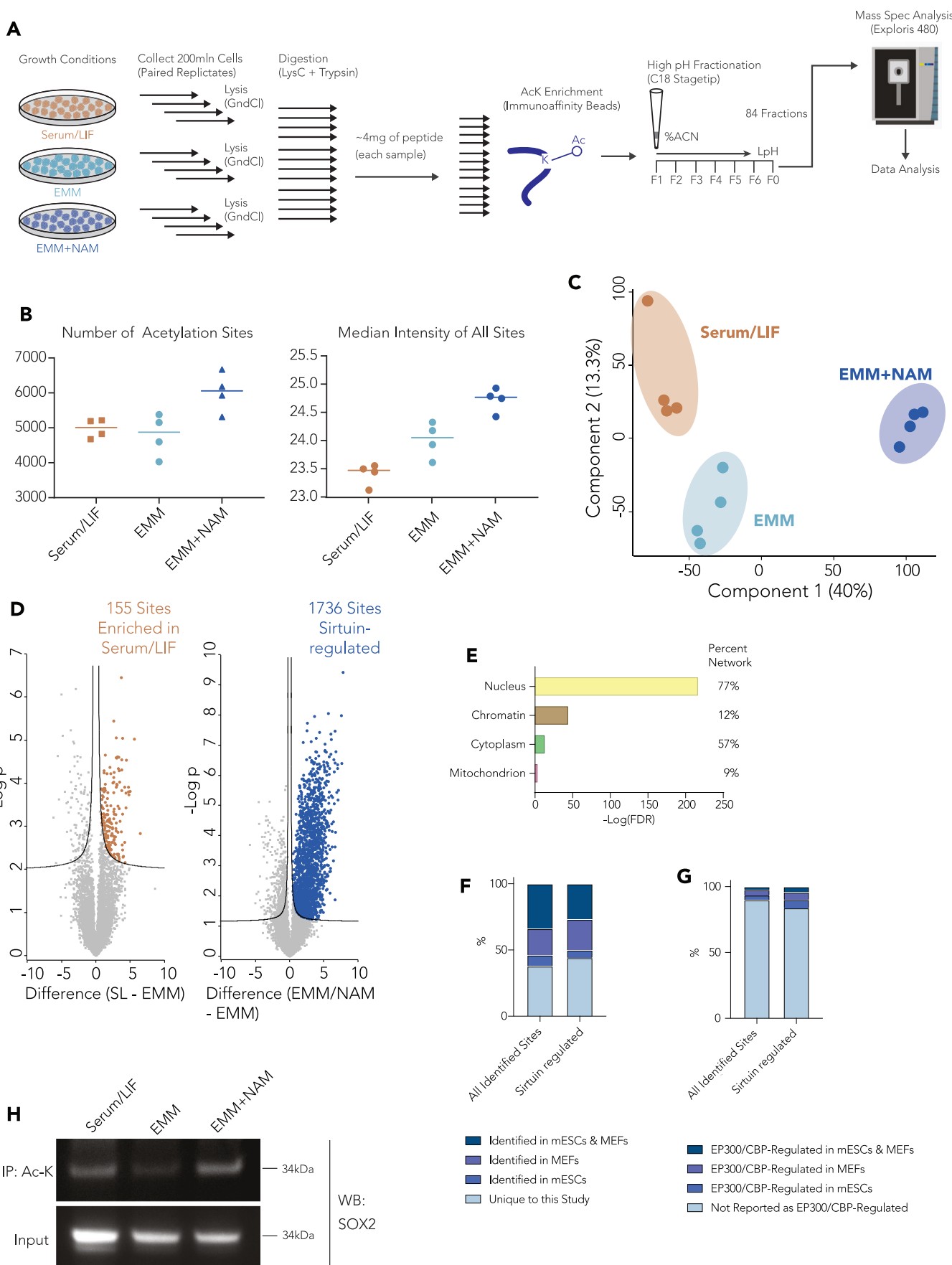

**Figure EV4.   Further characterization of acetylome of EMM-cultured ESCs.**

(A) The experimental setup to determine differential acetylation in ESCs cultured in Serum/LIF, EMM and EMM + NAM for 24 h. $n = 4$ experimentally independent samples. (B) Total number of acetylated sites detected in samples (left); Median intensity of all acetylated sites detected in samples (right). (C) PCA plot of acetylated sites in different samples. (D) Volcano plots depicting significantly differentially acetylated sites (FDR 5%) between EMM and Serum/LIF (left) and between EMM and EMM + NAM (right). (E) GO Cellular Component Enrichment table for sites deacetylated in EMM, with percent network. (F) Bar chart depicting percentage of all identified sites (6733) and that are also identified in datasets from (Weinert et al, 2018; Narita et al, 2021). (G) Bar chart depicting percentage of all identified sites (6733) and the overlap between the Sirtuin-regulated sites we defined (1731) and those regulated by EP300/CBP in datasets from (Weinert et al, 2018; Narita et al, 2021). (H) IP for Acetylated lysine and Western blot analysis of SOX2 in ESCs cultured for 24 h in Serum/LIF and EMM, $+/-$ NAM. Data are representative of 3 biologically independent samples.

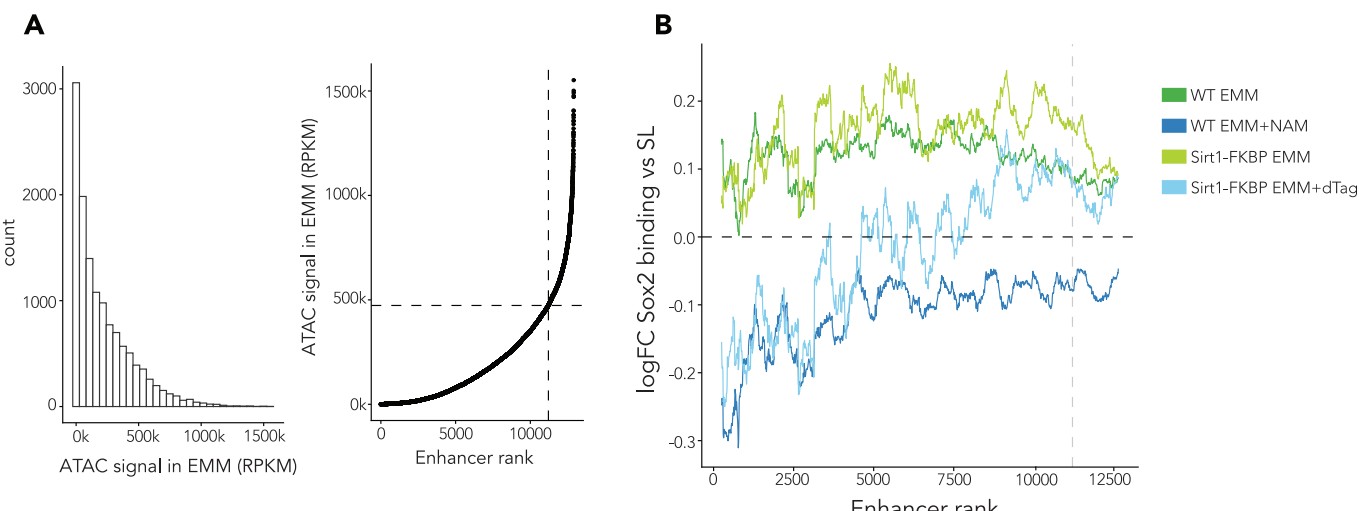

**Figure EV5. Extra analysis of SIRT1 deacetylation of SOX2, and of SOX2 CUT&Tag in Sirt1-FKBP cell lines.**

(A) Histogram ATAC signal count showing the number of SL-EMM merged enhancers plotted against ATAC signal in EMM (left), and enhancers ranked by ATAC signal (right). (B) Sliding window analysis of SOX2 signal pileup at enhancers ranked by ATAC signal in EMM, with the signal normalized to SL (for WT samples) and SL GFP+ (Sirt1-FKBP samples).

