## [Peer Review File · The EMBO Journal]

Altering metabolism programs cell identity via NAD⁺-dependent deacetylation

Robert Bone, Molly Lowndes, Silvia Raineri, Alba Redó Riveiro, Sarah Lundregan, Morten Dall, Karolina Sulek, Jose Romero, Luna Malzard, Sandra Koigi, Indra Heckenbach, Victor Solis-Mezarino, Moritz Völker-Albert, Catherine Vasilopoulou, Florian Meier, Ala Trusina, Matthias Mann, Michael Lund Nielsen, Jonas Treebak, and Joshua Brickman

Corresponding author: Joshua Brickman (joshua.brickman@sund.ku.dk)

Review Timeline:

Transfer Date:	18th Dec 24
Editorial Decision:	15th Jan 25
Revision Received:	3rd Feb 25
Accepted:	3rd Mar 25

Editor: Daniel Klimmeck

Transaction Report:

This manuscript was transferred to The EMBO Journal following peer review at another journal. The peer review comments and authors' responses were made available as agreed with the authors and the other journal, and were taken into account for the decision process at The EMBO Journal.

Response to Reviewers Comments for the previous journal

We wish to thank the reviewers for their very constructive input, our manuscript is much stronger having now incorporated their suggestions. We have responded to each of these comments below, but first we summarize some of the major new data that we feel has greatly enhanced our paper and provides fundamental insight into how global chromatin modifiers promote highly specific changes in gene expression by simultaneously reducing transcriptional or epigenetic noise, while at the same time stimulating lineage-specific signatures. We believe this concept of signal to noise regulation could have broad implications for how these factors act in aging.

Specific additions to our paper include:

1. Comparisons of EMESCs to 2i/LIF-cultured ESCs, addressing the similarities and differences between conventional defined naïve ESCs and EMESCs. This includes qPCR analysis of pluripotency genes: *Oct4*, *Sox2*, and *Tfcp2l1* alongside lineage-specific genes, *Gata4*, *Sox7*, *cMyc*, *Eomes*, and *Gata3*. To characterize the activity of developmentally important signaling pathways, we also assessed the activity of the following pathways by Western blot for phosphorylated β -catenin, YAP, GSK3 β , AKT and STAT3. We find substantial differences in the signaling state of 2i/LIF-cultured ESCs and EMESCs, most notably EMESCs exhibit substantial ERK activation, which is not present in 2i/LIF. Both our gene expression analysis and the activation of these two pathways complements the enhanced capacity for extra-embryonic differentiation of EMESCs relative to 2i/LIF-cultured ESCs. Finally, we performed RNAseq (see below) on moderate and high passage 2i/LIF and EMESCs to compare habituated cell culture states, and find that unlike high passage 2i/LIF cells that express high levels of apoptosis markers and are known to destabilize, EMESCs appear highly stable following prolonged culture (Extended Data Fig. 3g-i).
2. Comprehensive new studies on EMM mechanism. The most common concern of the reviewers was the lack of substantive mechanistic evidence for the role of Sirtuins (specifically SIRT1) in mediating the downstream response to metabolic change in EMM. We have addressed this concern in a variety of ways. First, by performing CUT&Tag analysis on transcription factors (TFs) identified in our combined acetylome/IMAGE analysis, and on histone marks H3K27ac and H3K4me1 to identify our own specific enhancer datasets in Serum/LIF, EMM and EMM+NAM (Fig. 3a-d and Extended Data Fig. 5). Based on this data, we conclude that the number of active enhancers is decreased in EMM (from approximately 14K to 4K), in agreement with new quantification of histone acetylation, that shows global reductions accompanied by an increase in methylation. However, the activity of the remaining EMM specific enhancers is increased based on levels of accessibility, acetylation, and transcription factor binding. Inspection of the deacetylated residues in the set of TFs we identified as present at EMM enhancers and modified in our acetylome (Fig. 2f) identified a single TF where acetylation/deacetylation occurs in the DNA binding domain, SOX2. Consistent with this observation, we see enhanced SOX2 binding at EMM regulated enhancers. While this is not observed for the other TFs we examined (KLF4 and TEAD1), the binding of these factors is specifically enriched at EMM enhancers where SOX2 binding is also enriched, suggesting deacetylation is driving cooperative binding by these lineage specific TFs. This

suggests that deacetylation acts simultaneously on TFs and chromatin, to decrease transcriptional noise and to stimulate lineage-specific transcription. Second, to demonstrate the ubiquitous nature of this model as a means for metabolic regulation of ‘lineage fitness’, we used conservation mass action kinetics to show that a simultaneous increase in TF and chromatin affinities creates enhanced specificity and TF dose response (Fig. 3e). Third, we generated SOX2 mutants in which the affected residues are mutated to mimic either an “acetylated” or “deacetylated” state and show that we can recapitulate EMM phenotypes in SOX2-Knockdown (KD) cell lines (Fig. 4a-c).

3. Another common issue for the reviewers was of inhibitor specificity in our experiments. This has been addressed by generating both a SIRT1 homozygous knock out (KO) ESC line, and a SIRT1-FKBP ESC line, in which SIRT1 can be rapidly degraded (Extended Data Fig. 9). We have repeated the analysis of SOX2 binding in the SIRT1-FKBP cells to illustrate the requirement of SIRT1 in the changes in enhancer activity and TF binding in EMESCs (Fig. 3f-h, Extended Data, Fig. 10).
4. We have addressed the point made by reviewer 2 on whether Fatty Acid Oxidation (FAO) is activated in EMESCs. We performed a C¹⁴ Palmitate Oxidation assay on Serum/LIF- and EMM-cultured ESCs, and we found that FAO is significantly upregulated in EMESCs compared with Serum/LIF-cultured ESCs (Extended Data, Fig 6h).
5. To address the questions of quality control in EMESCs, we performed blastocyst injection of EMESCs, and allowed the chimeras to develop to full term. We then assessed their contribution to the germ line and found that they contribute effectively. We also performed RNA-seq analysis on ESCs cultured for 10 and 20 passages in EMM, alongside ESC cultured in Serum/LIF and 2i/LF. We found that transcription of apoptosis genes was significantly lower in EMESCs compared to 2i/LIF-cultured ESCs, while levels of embryonic and extra-embryonic gene expression remain stable (Extended Data Fig. 3g-i). This demonstrates the stable quality of EMESCs, and the suitability of EMM for long term culture for ESCs.

We believe that the changes we have listed above have addressed the major concerns of all four reviewers and have strengthened the claims made in our manuscript. In particular we believe that our manuscript now provides compelling evidence for a new paradigm in chromatin regulation addressing how global changes in metabolism can affect highly specific alterations in gene expression, while damping down transcriptional noise. This effective means of increasing the transcriptional signal-to-noise ratio could be broadly relevant to development, differentiation, and aging.

REVIEWER COMMENTS

Reviewer #1 (Remarks to the Author):

In this study, Bone and colleagues consider how a forced shift in metabolism can influence cell fate. The authors drive mouse ESC into a metabolic program dominated by oxidative phosphorylation, the hypothesis being that this metabolic transition will convert the cells from a developmental state equivalent to pre-implantation epiblast to a phenotype nearer that of the inner cell mass (capable of forming extraembryonic tissue). The authors characterize the metabolically reprogrammed cells in terms of cell cycle status, prevalence of spontaneous apoptosis, developmental capacity, gene expression, metabolomics and chromatin status. From this extensive characterization, the authors argue that the shift in metabolism has produced a renewable cell population (called enhanced metabolic ESC or EMESC) that indeed resembles the inner cell mass of the pre-implantation mouse embryo. The authors link these metabolic changes with the activity of sirtuin histone deacetylases to account for much of the cellular reprogramming.

The transitions that cells undergo from the totipotent status of early blastomeres through to inner cell mass, pre-implantation and post-implantation epiblast, and up to gastrulation are of great current interest in the developmental biology and stem cell fields. Research into these transitions will enable us to better understand cultured cell populations in an embryonic context, and may help us to derive cultured stem cells with enhanced properties.

This study shows clearly that mouse pluripotent stem cells forced to rely on oxidative phosphorylation transition to a state that might be closer to the inner cell mass. This is the main point of the study, and it is clearly demonstrated. It is less clear how the new cell state relates to naïve cells resembling pre-implantation epiblast. Much of this study compares the EMESC to cells grown in serum/LIF. This latter culture condition does not maintain a pure population of pre-implantation epiblast-like cells; in fact, it is clear from the authors' own data (see below) that there is a considerable degree of spontaneous differentiation in these serum/LIF cultures. It would have been informative to compare EMESC to naïve cells grown in 2iLIF, since the latter are more equivalent to pure cultures of pre-implantation epiblast, which relies on both glycolysis and oxidative phosphorylation and has a restricted developmental potential relative to the inner cell mass (according to Extended Data Figure 3a, there is much greater reliance on OXPHOS in EMESC compared to 2i/LIF or KSOR/LIF, so why use serum/LIF as the baseline for the bulk of the work). This approach would distinguish this study more clearly from previously published work examining metabolomics of the primed to naïve pluripotency transition (such as for example the recent work of Cornacchia et al. in the human and many studies of mouse ESC), and would help establish unique features of the EMESC state.

- We thank the referee for suggesting these experiments. We have added data to specifically address the similarities of EMESCs with ESCs cultured in defined naïve conditions, namely 2iLIF. These include Western blots to examine signaling activation (new Extended Data Fig. 3a), RT-PCR for marker expression in Serum/LIF, 2i/LIF and EMM (new Extended Data Fig. 3b) and RNAseq on passage 10 and 20 culture in all three media conditions (extended data 3f-i). We believe this new data nicely distinguishes the EMESC state from naïve to primed transitions. In addition, we find our exploration of mechanism underpins far reaching conclusions

not anticipated in any of the studies exploring the metabolic basis for the naïve-to-primed transition.

Specific comments:

1. L 55-the authors should specify what they mean by totipotent, especially in light of recent papers including work from the Brickman laboratory showing that human naïve PSC can form extraembryonic endoderm and trophoblast.

- We have addressed this in the text. We recently discussed this issue at length^{1,2}, and we have tried to avoid extensive discussion of totipotency as it is loaded term.

2. L77-see note above relating to previous study of Cornacchia et al.

- We have addressed this in the text. This study, ref³, has been referenced on line 41 (reference 8).

3. L 117 and Ext data fig 3b-a very high proportion of cells in serum/LIF conditions express Gata6, is this degree of spontaneous differentiation expected. Likewise, 1 f shows a colony with very extensive differentiation and a high proportion of mixed colonies. I am slightly concerned with these controls.

- We have clarified in the text that the GATA6+ cells are very sparse in our culture medium, and there is a normal proportion of GATA6+ cells relative to other published Serum/LIF cultures.
- We have also reclassified the level of differentiation in the AP colonies in Fig. 1f (now new Extended Data Fig. 2e). As we now explain in the legend, we define two mixed categories: “mixed” category with “mixed <50% diff”; and “mixed >50% diff”. While the percentage of mixed colonies is high, normal Serum/LIF culture produces a majority of “mixed <50% diff” (new Extended Data Fig. 2e, lower panel) and therefore think it may be an issue of terminology.
- The levels of differentiation in these colonies as we define them are similar to those in previous publications⁴.

4. L 143 and following-the authors should have examined the ability of these chimeras to develop to term and contribute to the germ line. This is a stringent test of how normal these cells really are.

- We have addressed this concern by injecting EMESCs into blastocyst-staged embryos and allowing them to develop to term. Following birth, we found that they contribute effectively to the germ line (new Extended Data Fig. 3j).

5. L148 and following-similar contributions to extraembryonic lineages by 2i/LIF and EMESC cells raise questions about the biological differences between them.

- We used Western blot analysis to assess the differences in signaling in the Serum/LIF, EMM and 2i/LIF conditions. We found considerable differences between EMM and 2i/LIF. EMESCs have higher levels of both AKT and ERK phosphorylation (new Extended Data Fig. 3a). Levels of β -catenin activation are higher in 2i/LIF, while STAT3 signaling is similar in the two conditions.
- We also performed qPCR analysis for pluripotency and lineage-specific genes (new Extended Data Fig. 3b) and found that EMM does express similar albeit, slightly lower (not statistically significant) levels of certain pluripotency markers than 2i/LIF and higher levels of both endoderm and trophoblast markers.
- We have performed RNAseq on passage 10 and 20 ESCs grown either 2i/LIF and EMM (EMESCs). The differences between these cultures were found to be stable (i.e. pluripotency, and extra-embryonic gene expression) and EMM cultured cells express lower levels of apoptosis markers than observed in 2i/LIF.

6. L154 and throughout-from this point forward, the authors restrict their experiments to a comparison of EMESC with serum/LIF conditions. It might be more appropriate to ask how these new cells compare with 2i/LIF cells. Are they really very different, and if so, how? Specifically, can the authors clearly delineate these cells from naïve cells?

- See point 5 above. We have also stressed more in the text that the main reason why we compare to Serum/LIF is because the media are identical apart from the alterations to carbon source (e.g., glucose vs galactose), but this is now supported with more thorough analysis comparing 2i/LIF-cultured ESCs with EMESCs.
- We wish to emphasize that at a gene expression level, the major difference between 2i/LIF and EMM is the expression of extra-embryonic factors. EMM also expresses higher levels of pluripotency markers than Serum/LIF, but not 2i/LIF.

7. Figure 1g-are the differences between EMESC and 2i/LIF in terms of contributions to various lineages statistically significant? If so, how many biological replicates are represented?

- This experiment was performed on 3 separate dates, from the same H2B-tomato ESC line, and we have performed the two-tailed chi squared test to show that the differences between the media conditions is highly significant (Fig. 1f).

8. Figure 2c how were “mouse ESC” shown grown?

- Please refer to methods section, starting line 527.

9. L173-were there increased proportions of cells positive for PrEnd and trophoblast present in EMM cultures?

- We are not sure that we understand the question, as we never maintain that EMM cultures have trophoblast or PrE.

- EMM cultures are enriched for cells that coexpress PrE, Trophoblast and Epi (pluripotency) markers.
- EMM cultures are more efficient than conventional ESCs at differentiating to trophoblast and endoderm.

10. L178- this looks like a primed to naïve transition.

- While we have not compared the EMM transition to a primed to 2i/LIF transition, we have compared EMM to both Serum/LIF and 2i/LIF in our new analysis in Extended Data Fig. 3.
- As discussed above, EMM culture leads to enhanced ERK stimulation, which is the opposite of what would be expected for a naïve to primed transition (now discussed lines 138-142).
- The transition to EMM also involves upregulation of extra-embryonic gene expression, while retaining high levels of pluripotent gene expression.

11. Extended data Figure 6c-not sure this correlation is all that convincing- seems to be driven by upregulated genes.

- The reviewer is correct, it is not the strongest correlation and our data at the time was limited to ATAC (currently retained in manuscript as Extended Data Fig. 5d).
- To address this point, we have now performed CUT&Tag for two commonly used histone post translational modifications known to reflect enhancer activity (H3K27ac and H3K4me1), which provide a more accurate enhancer set for EMESCs.
- Based on the overlap of these two marks, we have defined enhancer sets for EMM, compared to Serum/LIF (new Fig. 5a).
- Based on the combination of these two enhancer datasets, a strong bidirectional correlation appears between the closing enhancers and reductions in acetylation and opening enhancers with increasing acetylation (new Fig. 3a, b; new Extended Data Fig. 5b, c).

12. L210-once again it seems to me these changes would also be seen in naïve cells v. serum/LIF

- We assume that the reviewer refers to the changes in enhancer accessibility in our two sets of PrE and pluripotency enhancers. At the time of the original submission, we had defined these based on ERK response, and it is unlikely that ERK responsive enhancers would come on during the transition from traditional primed to naïve transitions as the “naïve state,” has been defined based on ERK inhibition.

- We also observe a reduction in the number of active enhancers in EMM compared to serum (from 13705 to 3829). The EMM enhancers are more active (new Fig. 3a), but the overall number of enhancers is less. Based on our recent assessments of a defined condition similar to Serum/LIF (N2B27, activin, chiron and LIF) and 2i/LIF, the number of enhancers goes from 8869 NACL to 13699 in 2i/LIF². This would not fit with the notion that EMM simply represents a naïve to primed transition.

13. Extended data Fig 7a,b- Sorry, I have missed something here regarding the design of the experiment. What is the significance of the time points in the serum/LIF controls? What is driving the changes in the control cultures?

- We did this to detect the changes in metabolism that resulted from lack of glucose/pyruvate, rather than due to addition of fresh media. The time points in Serum/LIF are therefore intended to control for metabolic changes that occur as a result of replenishing the same media rather than changing to the new one. This has been clarified in the text.

14. L 258- please provide reference(s) regarding specificity of NAM as a sirtuin inhibitor.

- **Bitterman K.J., et al.,** "Inhibition of silencing and accelerated aging by nicotinamide, a putative negative regulator of yeast sir2 and human SIRT1." *J Biol Chem.* 2002; **277**: 45099-45107 (ref⁴). This reference has been added in the text.
- As referred to above, we have now validated our findings in *Sirt1* mutants (new Fig. 3f-h; new Extended Data Fig. 9 and 10a, b).

15. L264- same point as 12 above relating to Ex-527, then again for AK-1 and 3-TYP.

- The reference for Ex-527 is – Gertz et al., "Ex-527 inhibits Sirtuins by exploiting their unique NAD⁺-dependent deacetylation mechanism." (ref⁵) This reference has been added to the text.
- The inhibitors for SIRT2 and SIRT3 (AK-1 and 3-TYP respectively) have been replaced, since they both lack specificity and we felt they were no longer necessary given our findings with the *Sirt1* mutants (new Fig. 3f-h; new Extended Data Fig. 9 and 10a, b).

16. L371- not clear how anti-aging and regenerative properties of sirtuins relate to embryonic stem cells.

- In essence a level of randomness in histone post translational modifications that occurs with aging is thought to lead to transcriptional noise that results in phenotypic variation (see reviewed in ref⁶). We find that EMM counteracts this phenomenon, and we show it here in ESCs. However, the implication is that EMM could be useful for culture of a number of different cell types.
- We have expanded on this subject to provide additional clarity in the Discussion (line 519).

17. L421 following-here, as they conclude, the authors raise some very interesting “big picture” points that merit further discussion

- As discussed above and in our covering comments on this response, we have expanded on these subjects. We thank the reviewer for their enthusiasm.

18. L432-how many ESC lines were studied, and used in what experiments?

- At least 3 independent biological cell lines were used in all experiments. Exceptions to this were for the ATAC-seq experiment, in which 2 biological replicates were used.
- For the majority of the paper, we used E14 derived ESCs. Some of the chimera experiments used both E14s and then additionally two independently derived F1 hybrid ESC lines (129/S2;C57BL/6, see ref³).

Referee #2 (Remarks to the Author):

The manuscript covers an impressive range of techniques and data, and builds an interesting hypothesis. However, as presented, the data fail to support the conclusion that fatty acid oxidation and oxidative phosphorylation activate NAD dependent sirtuin de-acetylation to drive transcriptional activation.

- We thank the referee for their support.

1. The primary data supporting activation of fatty acid oxidation are transcriptional increases in some fatty acid oxidation genes. Curiously, the authors never test fatty acid oxidation directly. Instead they are using oxidative phosphorylation in the setting of full glucose media as a surrogate (seahorse assay). Later, the authors supplement the media with EMM, again as a surrogate for fatty acid metabolism. But never establish that fatty acid metabolism was altered. The authors circle back to fatty acid metabolism via transcriptional changes, but again never test this directly. This is a fatal flaw in their rationale.

- We thank the referee for pointing us to these experiments. To further explore the effects of EMM culture of ESCs on fatty acid oxidation, we performed a C¹⁴ Palmitate oxidation assay on ESCs cultured in Serum/LIF and EMM, then measured C¹⁴ CO₂ levels as a direct readout of FAO (new Extended Data Fig. 6h).
- We found that FAO was significantly upregulated in EMESCs relative to Serum/LIF-cultured cells, consistent with the increase in Cpt1a levels (Extended Data Fig. 6f, g).

2. In multiple instances, the authors show transcriptional changes, chromatin changes, and occupancy changes that go in both directions. But then conclude that one of the directions is slightly favored, thereby supporting their model. The lack of a strong signal and almost all large scale Omic changes weakens their conclusions.

- We have addressed this point by looking directly at chromatin and the occupancy of key TFs identified in our combined acetylome/IMAGE analysis, using CUT&Tag in Serum/LIF, EMM and EMM+NAM conditions (new Fig. 3b-d).

- We demonstrate that the number of active enhancers (based on H3K27ac and H3K4me1), is decreased in EMM (new Extended Data Fig. 5a, from 13705 to 3829), consistent with a general reduction in histone deacetylation (new Fig. 2d, new Extended Data Fig. 6j, k). However, at these select EMM activated enhancers we observe a robust increase in acetylation or activity (new Fig. 3a, new Extended Data Fig. 5e).
- To really focus on mechanism, we identified one transcription factor, SOX2 (new Fig. 4a), which is deacetylated in its DNA binding domain. We find that SOX2 binding is stimulated by EMM (new Fig. 3b, c), while the two other pluripotency transcription factors, KLF4 and TEAD1, did not exhibit the same general increase in binding. However, these factors were specifically targeted to enhancer regions with elevated SOX2 binding in EMM, suggesting that deacetylation drives cooperative recognition of key enhancers, but that this requires SOX2 co-binding.
- We test the putative role of increased histone binding (as a result of deacetylation) at the same time as creating a similar augmentation in select transcription factors, by mathematical modeling (new Fig. 3e). This model shows that manipulating both parameters (histone and TF affinity) simultaneously is predicted to lead to a highly specific, concentration dependent, transcriptional response.
- We also specifically test the role of SOX2 in this process by generating acetyl mimic mutations in SOX2 (new Fig. 4a). We replaced the 2 lysines identified in our acetylome, as well as one previously published residue⁷ and expressed these SOX2-mut constructs in a SOX2-FKBP cell line (a kind gift from the deWitt lab). Our findings show that the “deacetylated” mimic supports undifferentiated ESC colonies, while the “acetylated” mimic has very little capacity to support clonal growth, even in EMM (new Fig. 4b, c). This supports our hypothesis that the deacetylation of these residues in SOX2 is essential for establishing the EMM phenotype.

3. The authors never really show strong evidence for roles for sirtuins. Instead, the absence of a role for PARPs leads them to conclude the changes must be driven by sirtuins. Aside from SIRT1, the sirtuins do not have well established inhibitors *in vivo*. Therefore the effects of the compounds on SIRT2,3 are considered weak, and should be supplemented with genetic data (kd,ko) to confirm the role.

- We thank the reviewer for this suggestion, and we have now introduced extensive new analysis in both a Sirt1 KO cell line (new Extended Data Fig. 9a-d), and a SIRT1-FKBP cell line (new Fig. 3f-h; new Extended Data Fig. 9f-i).

Furthermore, the authors never really show sufficiency for the sirtuins. Instead, they are simply showing some increases in acetylation in the putative absence of sirtuin activity. Clearly a myriad of biological conditions can lead to increases in acetylation, and so a more direct role for this family of proteins should be tested and shown.

- As stated above, our inclusion of Sirt1 mutant cell lines, where we directly assess the role of SIRT1 in EMM-mediated deacetylation we believe addresses this concern.

Lastly, NAM is a weak inhibitor of sirtuins in vitro, and unfortunately can be taken up and reactivated by the salvage pathway to make more NAD in vivo, thereby confounding results. The authors should test whether or not NAM is activating the salvage pathway in their system.

- Again, while this is true, our demonstration that EMM has little effect in the absence of SIRT1 on both histone deacetylation (new Extended Data Fig. 9d, i) and SOX2 binding (new Fig. 3f-h), should provide reassurance here.
- Our new histone mass spec analysis also confirms that SIRT1 target residues are rescued from the EMM phenotype by the addition of NAM (new Fig 2d; new Extended Data Fig. 7d).

Referee #3 (Remarks to the Author):

This paper describes a series of experiments to understanding how metabolic capacities of cells reciprocally regulate gene networks and cell fate/identity. There are a lot of diverse observations that suggest balancing glycolysis and OXPHOS, NAD metabolism, sirtuins and select transcription factors are important. Altogether, this is interesting, however, the take home message is rather vague, the mechanistic parts are not very compelling (relying too heavily on pharmacological agents, and the idea that metabolism is linked to cell fate is important, but not new. While the work has potential, in its current form, the paper feels premature and incomplete. Many of the mechanistic experiments are not addressing causation, but instead, suggest correlation, and therefore will require more definitive experiments.

- We thank the referee for their generally supportive comments. We feel that we have addressed them on several levels (see above) and briefly reiterated below. Also see our specific responses to reviewer 1 and 2.
- We have honed the take home message to the role in metabolically driven histone deacetylation in increasing the transcriptional signal to noise ratio, by enhance TF specificity via deacetylation and reducing the number of available non-specific sites via histone deacetylation.
- We have addressed our reliance on pharmacological regents by adding Sirt1 and Sox2 mutants to our analysis.
- We tested the hypothesis that metabolic change can be transmitted into a specific transcriptional response via the simultaneous inhibition of both histone and TF acetylation, by generating a computational model, assessing TF occupancy in a SIRT1-degradable ESC line, and generating acetylation mimics and mutants in Sox2-FKBP cells.

Referee #4 (Remarks to the Author):

The manuscript by Bone et al demonstrate that by replacing D-glucose and pyruvate with D-EMM in regular serum/LIF medium, mouse ESCs can be metabolically reprogrammed to a more ICM-like state, dubbed enhanced metabolic ESCs (EMESCs), by forcing enhanced dependence on OXPHOS at the expense of glycolysis. They further demonstrate that enhanced activities of Sirtuin family deacetylases in the deacetylation of ICM-specific TFs are responsible for enhanced pluripotency of EMESCs through re-shaping the EMESCs-specific gene network and chromatin accessibility.

Conceptual Novelty: Limited. It is noteworthy that there is no actual mechanism presented in this study. Almost all informatics data are largely descriptive, and merely based on in vitro studies of ESCs. Given that the metabolic switch between OXPHOS and glycolysis during naïve/primed pluripotent states are well known, the conceptual novelty of this study is quite limited.

- We disagree with this as even the first version of our manuscript contained the outlines of the mechanism, that deacetylation acts in parallel on TFs and chromatin to produce a highly specific response. However, the new manuscripts both contains this explicit hypothesis and then tested it in both mutants and via a computational modeling approach.
- In our response to reviewer 1 we explained how the shift from standard culture to EMM is not a “primed to naïve,” transition, but rather the induction of a more ICM like state. Essentially EMM does not create a naïve 2i/LIF-like state at the level of signaling (new Extended Data Fig. 3a) or transcriptionally (new Extended Data Fig. 3b). In particular, EMM induces a “naïve state”, in which ERK is robustly active.
- EMM also does not induce a new “naïve,” enhancer set, but rather fine tunes the potential enhancer set from 13705 to 3829 active enhancers (new Extended Data Fig. 5a) and this is not what we observe generally when primed and naïve enhancer sets are compared².

Significance: Although EMESCs illustrate coexpression of ICM and PrE reporters, a phenomenon of ICM-like state, however, it is not clear if EMESCs are genetically stable, or if the EMESCs can be reprogramed or differentiate to multiple lineages properly compared to regular serum/LIF or 2i/LIF ESCs. The advantage for future applications of EMESCs is also unclear.

- Our new manuscript shows that EMESCs can produce both high level born chimeras and contribute to the germ line (new Extended Data Fig. 3j). We believe that this should be sufficient to demonstrate that cells cultured in EMM are genetically stable and maintain a normal karyotype. We have also performed RNA-seq analysis on EMESCs cultured for 10 and 20 passages and found that levels of apoptosis genes remain low compared to 2i/LIF-cultured ESCs at those passages (new Extended Data Fig. 3g). Long term EMESCs also display, stable expression of embryonic and extra-embryonic genes (new Extended Data Fig. 3h, i). Together, these data demonstrate the suitability of EMM as a culture media for culturing of stable ESCs with enhanced function. Moreover, 2i/LIF is generally not used for gene targeting as cells crash at

high passage numbers, and we show that EMM avoids the elevated levels of apoptosis markers induced by many passages in 2i/LIF.

- We also believe that our significance lies not in the cells themselves, but in the mechanism by which EMM reprograms cells. We feel that our new mechanistic studies suggest that our findings have broad implications, since they demonstrate how metabolically driven chromatin-modifying enzymes can have a highly specific influence on transcription, as they increase signal to noise ratios at the level of lineage specific transcription. Moreover, this mechanism explains how these enzymes can counteract the enhanced levels of “epigenetic noise,” that underlies phenotype variation resulting from aging.

Overall, the work is interesting, although the conclusions drawn and the mechanism dissected are weakly supported by the experimental data provided. The techniques used in this study, the depth in which the mechanism of the process is studied and the validation of the quality of the EMESCs do not add much to the existing knowledge in the field.

- We thank the reviewer for these comments. Although we disagree on the novelty and importance of this work, we also recognize the important points the reviewer makes regarding the need for more mechanistic experiments, which we believe we have addressed in the revised manuscript.

Major critiques:

1. Almost all informatics data are largely descriptive and based on illusive Go and KEGG pathway analysis of hundreds of DEGs. It is hard to appreciate if their conclusions are solid enough. In addition, heatmaps (Fig 2d, S5c) showing expression differences between different groups of lineage genes were somehow cherry-picking, it is not clear how the metabolic switch can affect the global gene expression of EMESCs.

- While there is an extent that any set of marker analysis can be viewed as cherry picking, we feel that we made quite a good sampling of each category in these lineages. Our heatmaps are there to show general trends of categories of genes in EMESCs, but these are merely illustrative examples. Below we summarize some of the analysis we have done to derive our hypotheses from these large datasets in a systematic and unbiased fashion.
- We have now added a comparison of EMM to 2i/LIF, at the level of signaling (new Extended Data Fig. 3a) and gene expression (new Extended Data Fig. 3b and g-i), demonstrating that EMM is not merely promoting a naïve state. In particular we identify active pathways known to be important for the PrE and TE lineages (ERK and Hippo respectively), which substantively add to our general understanding of cell state induced by EMM.
- As part of our effort to decipher the mechanism we performed CUT&Tag on deacetylated TFs and histone post translational modifications (PTMs) (H3K27ac and H3K4me1) to aid in the definition of enhancers (EMM-responsive enhancers, new Extended Data Fig. 5a; TFs in wild-type ESCs, new Fig. 3b-d; TFs in the SIRT1-FKBP ESCs, new Fig. 3f-h and new Extended Data Fig. 10b).

- To support the findings from our metabolomics dataset, we also include extensive histone PTM analysis by mass spectrophotometry to quantify overall levels of a range of PTMs (see new Fig. 2d, new Extended Data Fig. 7d, e) and the above analysis of EMM-specific enhancers (see new Fig. 3a-d; new Extended Data Fig. 5a, e-g). We believe this new analysis provides a comprehensive and concrete description of the effect of EMM on both the epigenome and its specific influence on the gene regulatory network induced by EMM.
- We generated Sirt1-KO and inducible SIRT1-degradation ESC lines to demonstrate that the deacetylation of both chromatin and TFs is a direct result of SIRT1 activity (new Fig. 3f-h, new Extended Data Fig. 9, 10a, b).
- To probe the generality of our proposed mechanism, we generated a computational model to test how the simultaneous alterations in histone and TF affinities would affect binding. We demonstrated that the model predicts the simultaneous manipulation of both affinities would promote specificity (new Fig. 3e).
- Finally and most importantly, we integrated multiple omics datasets (metabolomics, acetylomics, transcriptomics and epigenomics) to focus on a set of TFs, in which only one, SOX2, was deacetylated in its DNA binding domain. We then generated Sox2 acetylation mutants and demonstrate they mimic aspects of the EMM phenotype (new Fig. 4a-c). Of the TFs we tested by CUT&Tag, only SOX2 had augmented binding in EMM to the enhancers we defined here. The other factors we tested, KLF4 and TEAD1, had enhanced binding to regions where their sites overlapped with SOX2, but not elsewhere. Taken together, through a careful and extensive assessment of this data we arrived at the hypothesis that the deacetylation of SOX2 by SIRT1 in EMM enhances its affinity to specific enhancers and supports cooperative binding with a set of deacetylated TFs that can compete effectively with higher affinity chromatin.

2. The quality assessment of EMESCs could have been addressed in much detailed manner.

- We have addressed this concern by injecting EMESCs into blastocyst-staged embryos and allowing them to develop to term. Following birth, we analyzed their ability to contribute the germ line. (new Extended Data Fig. 3a, b)
- We have also compared EMESCs to 2i/LIF-cultured ESCs over many passages (new Extended Data Fig. 3f-i). The decreased levels of apoptosis genes in EMESCs compared to 2i/LIF-cultured ESCs may be because 2i/LIF-cultured ESCs are grown in the presence of a continual block to ERK signaling, and therefore we believe EMESCs are an attractive alternative model for pluripotent stem cell culture.
- Finally, while we agree with the reviewer that this is an important issue, we think our mechanistic studies described above are where the underlying significance comes from.

3. The genomics and proteomics experiment have been superficially used in the manuscript and only selected examples have been cherry-picked to draw conclusions.

- We hope that our new data and analysis provides more reassurance. In our response to point 1 we go into considerable detail about how we have both incorporated new data and performed global analysis that supports our hypothesis. We hope this satisfies the reviewer that we have not cherry-picked anything.

4. In Fig 2c: The PCA alone is not enough to support the claim that the expression profiles of EMMS (3-24h) is close to ICMs. This way they are also equidistant to 2-cell stage.

- This is true, however when we analyze the expression of 2c genes in EMESCs (current Extended Data Fig. 4e), we note that there is no significant change in expression. However, in this analysis we observe a marked increase in the expression in the genes expressed in ICM-stage embryos.
- GO analysis of the genes in the vicinity of EMM regulated enhancers (new Extended Data Fig. 5b), does not highlight repeat expression or 2c states, but does highlight LIF, blastocyst formation, and two trophoblast related terms.

5. From the untargeted metabolomics, the authors found out differentially enriched metabolites. It would be desirable to understand how these metabolites might be regulating the slow proliferation rate (48h) of the new cell types.

- We agree, but we believe this is beyond the scope of the manuscript and any speculation would be just that.

6. From Fig. 2e,f of ATAC-seq analysis during the metabolic switch, there is no picture of the global change of chromatin accessibility during this transition. In addition, the differences of ATAC-signals at pERK up-/down-regulated enhancers are trivial.

- In the previous version of the paper, we did include some global analysis (current Extended Data Fig. 4j, k).
- We agree with the reviewer that change in ATAC-seq at ERK regulated enhancers was not dramatic. To better assess how EMM was influencing chromatin at enhancers, we defined a set of enhancers using CUT&Tag for the histone marks, H3K27ac and H3K4me1. We then assessed EMM-influenced acetylation at these enhancers and subsetted them based on ATAC-seq (opening and closing, new Fig. 2f, new Extended Data Fig. 5c), then defined motifs at open and closed enhancers (new Extended Data Fig. 5d). We have now moved the analysis of the ERK regulated enhancers all to extended data. We hope that the reviewer finds the robust changes we observe in acetylation at regulated enhancers as convincing as we do.

7. Most of the functional work on Sirtuin family of proteins are based on small molecular inhibitors. It's not clear which Sirt protein (maybe Sirt1) is responsible for regulating the pluripotency TFs in EMESCs. Additional KD/KO work is required to further dissect the mechanism behind.

- We agree and have added a significant amount of work assessing the Sirt1 mutants and demonstrating that it is downstream of EMM (see above).

8. Some of their immunostaining pictures only showed a single positive cell in the field (Fig S3b, S7i). More representative images are required.

- This is true as the single-cell images referred to are Gata6+ cells, in standard ESC culture which are relatively rare.
- The point of this analysis is that in Serum/LIF these cells are GATA6 single positive, whereas in EMM they also co-express pluripotency TFs, but not in Serum/LIF.

9. The model is weakly supported by cherry picked examples only. It lacks substantial data support. .

- While our model was supported by candidate-based experiments previously, the current data including mutants, conditional mutants, refined enhancer datasets, and computational modelling robustly support our novel views of how metabolic regulation effects specific transcriptional changes and refines cell type specification.

Minor comments:

1. The authors used IMAGE to integrate their omics datasets and came up with a list of TFs that could be potential causal TFs for this process. It would be interesting to know if this alternative state could be achieved by reprogramming done with these TFs.

- This is an interesting point; however, we feel that addressing this would be outside the scope of this paper.

2. What's the reason for picking Spry4 in 2e and not all 3 representative genes from each category mentioned in 2d?

- We are a bit confused by this comment, does the reviewer want three genes from each category? Or is the question directed at our PrE marker set that doesn't include Spry4. Spry4 was chosen based on it having a super enhancer that responds to ERK and it being a marker of PrE⁸.
- Spry4 wasn't included in the set of marker genes as it is less familiar as an endoderm marker than the others indicated on the heat map (Extended Data, Fig. 4e).

3. Line 70: which “all three conditions” are not clear from the text unless one checks out the supplementary file

- We have changed this in the text to make it clearer.

4. Line 18: “A cells mode” should be “A cell's mode”; Line 26, “A cells choice” should be “A cell's choice”.

5. Media (plural) instead of medium (singular) has been used at multiple places throughout the manuscript.

- We have made these corrections in the text.

REFERENCES

1. Riveiro, A. R. & Brickman, J. M. From pluripotency to totipotency: an experimentalist's guide to cellular potency. *Development* **147**, (2020).
2. Redo-Riveiro, A. et al. Transcription Factor Co-Expression Mediates Lineage Priming for Embryonic and Extra-Embryonic Differentiation. *Stem Cell Rep.*
3. Cornacchia, D. et al. Lipid Deprivation Induces a Stable, Naive-to-Primed Intermediate State of Pluripotency in Human PSCs. *Cell Stem Cell* **25**, 120-136.e10 (2019).
4. Bitterman K.J., et al., "Inhibition of silencing and accelerated aging by nicotinamide, a putative negative regulator of yeast sir2 and human SIRT1." *J Biol Chem.* **277**, 45099-45107 (2002).
5. Gertz, M. et al. Ex-527 inhibits Sirtuins by exploiting their unique NAD⁺-dependent deacetylation mechanism. *Proc Natl Acad Sci USA.* **110**, E2772-81 (2013).
6. Bartz, J. et al. Progress in Discovering Transcriptional Noise in Aging. *Int. J. Mol. Sci.* **24**, 3701 (2023).
7. Baltus, G.A. et al. Acetylation of Sox2 Induces its Nuclear Export in Embryonic Stem Cells. *Stem Cells.* **27**, 2175-2184 (2009).
8. Nowotschin, S. et al. The emergent landscape of the mouse gut endoderm at single-cell resolution. *Nature* **569**, 361–367 (2019).

Response to reviewers' recommendations for the previous journal

We respond below to the remaining reviewers' comments (red text). While we believe we have already extensively revised this manuscript experimentally we provide a few additional pieces of data to comprehensively rebut reviewer 4's additional critiques.

Reviewers' recommendations:

Reviewer #1 (Remarks to the Author):

In their revision, the authors have gone to considerable lengths to address critical concerns raised in the previous review. Specifically they offer new data comparing EMESC to naive cells, provide further analysis of chromatin status in EMM, include findings on Sirt1 knockout and degron lines, and undertake more rigorous study of the developmental potential of EMESC. These new data have greatly strengthened the manuscript and provide stronger support for its conclusions. The findings on the EMESC state will be of interest to developmental biologists and to stem cell biologists who may wish to examine the utility of these cells in a variety of applications. It will also be interesting to see if the authors' wide ranging speculations regarding aging turn out to be correct.

We would like to thank the reviewer for their support.

Reviewer #4 (Remarks to the Author):

Overall Assessment: While the authors have made revisions to address previous comments, significant concerns remain regarding the novelty and significance of the study. The relationship between Sirtuin and SOX2 in the context of pluripotency is well established, and the metabolic switch between oxidative phosphorylation (OXPHOS) and glycolysis in determining cell fate is also a thoroughly studied area. The manuscript fails to address these issues adequately, limiting the overall conceptual novelty of the work.

- While we agree that the role of SIRT1 in deacetylating SOX2 has been reported, our study suggests a new paradigm for metabolic regulation: the simultaneous deacetylation of chromatin and TFs to both decrease the non-specific binding and enhance specific binding. We believe this concept of metabolic mediated enhanced transcriptional signal-to-noise could have broad implications for development and aging.
- The observation that this is mediated through the DNA binding of a single TF, alongside deacetylation mediated cooperativity is also novel and fits the implied role of SOX2 in aging.

- As reviewer one points out, these “It will also be interesting to see if the authors' wide ranging speculations regarding aging turn out to be correct.” To us this implies that the paradigm we suggest has broad implications that have yet to be explored.
- Finally, the idea that ESCs, and perhaps other stem cells, can be cultured in galactose and avoiding multiple small molecules could represent a major new cell culture innovation. We have mentioned in a our conflict of interest statement that we have filed a patent in this area, we are also developing this jointly with a media provider/company. This is particularly interesting as galactose was recently shown to be a key factor in inducing diapause in human cells¹. We have refrained from mentioning this in previous correspondence with yourself as these activities have only begun recently.

Specific Comments:

1. Lines 20 and 28: There are typographical errors that have not been corrected, despite the authors' claim that these were addressed. The correct phrases should be "A cell's mode" and "a cell's choice" respectively. It is unclear why these basic corrections were overlooked.

This has been corrected.

2. Line 135: The sentence "...are much higher EMM" is incomplete and missing a crucial word. This needs to be rectified for clarity.

This has been corrected.

3. Fig. 2A: The blue font used is too faint and difficult to read. This should be corrected to ensure all figure elements are clearly visible.

This has been corrected.

4. Lines 359-360: The reference to Extended Data Fig. 8I is problematic as this figure panel is missing from the manuscript. Additionally, the relationship between Ext Data Fig. 8H and the statement made in the manuscript seems contradictory, causing confusion in interpreting the results.

This has been corrected.

5. Fig. 3A: This figure is mentioned out of order in the results section, which disrupts the flow of the manuscript. The authors should ensure that the figures are referenced in a logical sequence that aligns with the narrative.

This has been corrected.

6. Line 375: The sentence font should be bold as this is a section title, consistent with other sections in the manuscript.

This has been corrected.

7. Line 439-440: The statement "...that of reducing transcriptional noise while elevating specific signals, could underly how these factors attenuate the aging process" is unclear. The word "underly" is particularly confusing and should be revised for clarity.

This has been corrected.

8. Line 470: The claim that "EMESCs exhibit enhanced contribution to chimeras and extra-embryonic differentiation" is not supported by robust data. The low number of pups/litters tested (8 vs. 7) and the observation of only one chimeric pup born from EMM/Galactose-cultured Line 3.4 compared to Glucose-cultured Line 3.4 are insufficient to support such a strong conclusion.

We feel this comment is incorrect. We have tested two clonal lines, one that no longer contributes to chimeras. In this instance, it is true that we only got a single chimera in seven pups, but it was judged to be 100% ESC derived. We believe that by the standards of the field, that take into account the time and cost required for animal experiments, these are robust experiments.

COMMENTS ON THE AUTHORS' RESPONSE TO REVIEWER #2

1. Fatty Acid Oxidation (FAO) Testing: Referee #2 raises concerns about the lack of direct FAO tests, which could undermine the manuscript's main hypothesis. The authors have tried to address this with a C14 Palmitate oxidation assay (new Extended Data Fig. 6h). While this is a step forward, the link between increased FAO and the activation of NAD-dependent sirtuins remains somewhat indirect. The manuscript could have been stronger by directly showing that increased FAO drives NAD⁺ production or changes in sirtuin activity, connecting the metabolic shift more clearly to the transcriptional activation mechanism proposed. Remaining Gaps in my mind: While the C14-palmitate oxidation assay provides a good measure of FAO, what's missing is a stronger mechanistic link between FAO and sirtuin activation. The manuscript infers that FAO leads to increased NAD⁺ levels and sirtuin activation, but it doesn't directly show that the increased FAO is responsible for NAD⁺ production or that this drives sirtuin activity. Direct evidence could include: 1) Measurement of NAD⁺/NADH ratios in the context of FAO activation. 2) Demonstrating that manipulating FAO directly (e.g., through genetic or pharmacological inhibition of CPT1A) affects NAD⁺ levels or sirtuin activity.

- We thank the reviewer for this thoughtful comment. However, we thought that we had sufficiently addressed Reviewer 2's primary concern, which was to demonstrate that lipid metabolism, in particular fatty acid oxidation (FAO), was increased in EMESCs. Combined with our original data showing increased FAO gene and protein expression (Supp. Fig. 6F, G), we show conclusive evidence of increased FAO in EMM (Supp. Fig. 6H).
- Although the combination of Extended data Fig. 6H and I, shows that increased FAO is accompanied by increased SIRT1 activity, we agree with the reviewer that a functional link between FAO metabolism and EMM phenotypes would strengthen our claims. To address this, we generated mutant ESCs in the major rate limiting fatty acid transporter, CPT1A. The figure below (Review Response Fig. 1) shows the phenotype of homozygous *Cpt1a* knock out (-/-) ESCs in self-renewal in either EMM or standard serum/LIF media. *Cpt1a* -/- ESCs form fewer colonies and fail to establish the EMESC phenotype based alkaline phosphatase staining and morphology, demonstrating that FAO is required for EMM activity. We believe this genetic demonstration that the EMM phenotype depends on FAO is stronger than the pharmacological inhibition of CPT1A as suggested by the reviewer in their point 2 above.

Review Response Figure 1: EMM colony phenotype is lost in *Cpt1a* (-/-) mutant ESCs. Cells were plated at clonal density, expanded for nine days and then stained for alkaline phosphatase.

2. Chromatin and Transcription Factor Analysis: The authors attempt to address the comment about weak signals from large-scale omic data by focusing on select

transcription factors like SOX2 (new Fig. 3b-d) and testing deacetylation's role via mutation studies. This targeted approach strengthens their claim by providing mechanistic insights into how SOX2's deacetylation regulates enhancer activity in response to the EMM condition. However, the conclusion about other pluripotency factors, such as KLF4 and TEAD1, could seem speculative without deeper follow-up. Furthermore, despite some improved clarity, the weakness of large-scale signals mentioned by the reviewer may still not be fully resolved, potentially limiting the broader impact.

- While we agree the original signal in our omics was subtle, we believe our current findings are robust, demonstrating extremely convincing signals with quantitative alterations in enhancer activity and binding of SOX2, KLF4 and TEAD1.
- Our targeted approach to these TFs was guided by the results in our original datasets. We identified SOX2 as the sole factor in our IMAGE/acetylome analysis which contained SIRT1-mediated acetylation/deacetylation in its DNA-binding domain. As our proteome and genome wide data identified a single TF in this class, it would appear a very strong signal, as opposed to a weak one. Furthermore, this striking observation was validated, as the other TFs we identified do not exhibit the same deacetylation dependent behavior as SOX2.
- The reviewer suggests that “the conclusion about other pluripotency factors, such as KLF4 and TEAD1, could seem speculative without deeper follow-up. What “follow up,” is required? We have already shown their binding is influenced by EMM (Fig. 3b, c, and d) at enhancers that are also occupied by SOX2, and the reviewer does not specify how they would like us to address the role of these factors further.
- We can also show that EMM-mediated deacetylation enhances SOX2 and TEAD4 interactions (see Review Response Fig. 2) but believe that further follow up on the nature of these complexes in EMM is beyond the scope of this manuscript.

Review Response Figure 2: SOX2 and TEAD4 co-binding is dependent on Sirt1 activity in EMM.

3. Role of Sirtuins: A significant point of contention was the insufficient evidence of sirtuins' role in the observed effects. The authors respond by introducing Sirt1 KO and FKBP cell lines, showing that SIRT1 is necessary for the deacetylation and SOX2 binding (new Fig. 3f-h). This addition is a strong point in their favor, directly addressing the reviewer's request for genetic validation. However, the reviewer also points out that the manuscript lacks demonstration of sufficiency for sirtuins, and this concern remains partially unaddressed. A more compelling case could have been made by showing that activating sirtuins alone recapitulates the effects seen with EMM.

- We appreciate the reviewer's comments, especially in recognizing the considerable effort to address their original concerns. However, we disagree that we have shown insufficient evidence as to the specificity of Sirtuins in EMM. We demonstrate that SIRT1 is required for TF binding to EMM enhancers with new CUT&Tag data for SOX2 in Sirt1-FKBP ESCs (Fig. 3F-H). We show in our histone mass spec data (Fig. 2D, Supp Fig. 7D) widespread deacetylation of histones, particularly the SIRT1 target H4K16ac. Furthermore, we demonstrated that SIRT1 is specifically required for histone and SOX2 deacetylation in EMM (Supp Fig. 9D-E). Therefore, we believe that we have shown that the major phenotypic changes caused by EMM are mediated by SIRT1.
- The sufficiency experiment requested by the reviewer is difficult, as SIRT1 overexpression won't necessarily provide a gain-of-function, as the enzyme requires activation by NAD. Overexpression can also be difficult to interpret as the levels

of SIRT1 would not be physiological. We believe that the gold standard for protein function is a genetic loss of function with rescue, which is essentially what we demonstrate in our rescue of *Sirt1*^{-/-} ESCs with a rtTA inducible degradable SIRT1.

4. Concerns with NAM: The critique of nicotinamide (NAM) as a weak sirtuin inhibitor and its reactivation by the salvage pathway presents a substantial challenge. The authors' response, while addressing this through histone mass spec data (new Fig 2d) and *Sirt1* KO cells, may still be viewed as insufficient to fully remove the confounding effects of NAM. A more rigorous inhibitor or independent system to demonstrate sirtuin activity might have been a better approach.

- While we acknowledge that NAM was not sufficient to show EMM depended on Sirtuins, we have provided an inducible Sirtuin mutant (see response to previous comment) and demonstrated that EMM activities depend on SIRT1. Specifically, EMM-enhanced binding of SOX2 to target enhancers depends on the presence SIRT1. We also found that NAM and SIRT1 depletion have the same effect in this context (Fig 3G, H).
- The recycling of NAM in the NAD salvage pathway is well documented, and we are aware that there are conflicting studies suggesting that NAM can either inhibit or activate Sirtuin activity, depending on the context. However, in the context of EMM culture, we clearly show that NAM inhibits Sirtuin activity. Based on quantitative histone mass spec data (Fig. 2D, Supp Fig. 7D) we show widespread histone deacetylation in EMM, particularly the SIRT1 target H4K16ac. All SIRT1-dependent deacetylations are inhibited by NAM. Additionally, NAM inhibits SIRT1-dependent SOX2 deacetylation and enhanced DNA binding at EMM enhancers (Fig. 3F-H). Together, we believe our data show that NAM inhibits SIRT1 activity, which is responsible for the EMM phenotype.
- We have additional data in the paper testing the SIRT1 inhibitor (Ex-527), P300 (A-485), and PARP1 (Olaparib) in extended data Figure 7.

My overall impression:

While the authors' revisions address many of the technical critiques raised by Referee #2, significant gaps in novelty and mechanistic depth remain. The relationship between SIRT1, SOX2, and pluripotency is already well-established in the literature, and the added data may not contribute enough new understanding to push the field forward. Similarly, the role of metabolism in stem cell fate determination (OXPHOS vs. glycolysis) is a well-explored area, limiting the impact of these findings without a stronger mechanistic or conceptual breakthrough. In its current state, the manuscript does not meet the high novelty and conceptual standards typically required by this journal for publication. However, it could still be competitive in a more specialized journal that focuses on stem cell metabolism or chromatin regulation.

We disagree that the significance of this study is limited – it is of course true that there are many studies investigating the role of metabolism in stem cell fate determination, however

we have uncovered a key link between ESC rejuvenation and reduced epigenetic noise, caused by a metabolic shift. Moreover, epigenetic noise that has been linked to aging was previously thought to be based solely on background levels of non-specific transcription. In contrast, our findings suggest that it is about transcriptional signal-to-noise ratios, triggered by the simultaneous activity of epigenetic enzymes on both TFs and chromatin. The reviewer has made statements about the lack of broad significance of our work, but has not commented on this observation, which we believe represents a new paradigm for epigenetic modification and could underlie how acetylases and deacetylases can illicit very specific cellular response, despite the previously described general activities of these enzymes on chromatin. As a result, we believe our study has far reaching implications for the aging field, provides insight into biological specificity and defines a new simplified stem cell state that could be linked to diapause.

References:

1. Iyer, D.P., Khoei, H.H., Weijden, V.A. van der, Kagawa, H., Pradhan, S.J., Novatchkova, M., McCarthy, A., Rayon, T., Simon, C.S., Dunkel, I., et al. (2024). mTOR activity paces human blastocyst stage developmental progression. *Cell* *187*, 6566-6583.e22. <https://doi.org/10.1016/j.cell.2024.08.048>.
2. Nishida, T., Naguro, I., and Ichijo, H. (2022). NAMPT-dependent NAD⁺ salvage is crucial for the decision between apoptotic and necrotic cell death under oxidative stress. *Cell Death Discov.* *8*, 1–11. <https://doi.org/10.1038/s41420-022-01007-3>.
3. Hasmann, M., and Schemainda, I. (2003). FK866, a Highly Specific Noncompetitive Inhibitor of Nicotinamide Phosphoribosyltransferase, Represents a Novel Mechanism for Induction of Tumor Cell Apoptosis. *Cancer Research* *63*, 7436–7442.

Dear Josh,

Thank you again for the submission of your amended manuscript (EMBOJ-2024-119954-T) to The EMBO Journal. Please accept again my apologies for the unusual protraction due to delayed expert input. We have carefully assessed your manuscript and the point-by-point response provided to the referee concerns that were raised during review at a different journal. In addition, and as mentioned before, we decided to involve previous referee #1 as an arbitrating expert to evaluate the revised version of your work, with respect to technical robustness, conceptual advance and overall suitability of your work for publication in The EMBO Journal.

As you will see from his/her comment enclosed below, the advisor is in broadly favour of the work stating the interest and value of your results and s/he is supportive of publication at The EMBO Journal.

We are thus pleased to inform you that we can offer to swiftly move forward towards acceptance of this work at The EMBO Journal.

We now need you to take care of a number of minor issues related to formatting and data annotation, which I will share shortly in a separate message, together with additional changes and requests by our production team and my colleague H. Sonntag (CC'ed) for Source Data provision.

Please submit a revised version of the manuscript using the link enclosed below, addressing the advisor's comments.

As you might have seen on our web page, every paper at the EMBO Journal now includes a 'Synopsis', displayed on the html and freely accessible to all readers. The synopsis includes a 'model' figure as well as 2-5 one-short-sentence bullet points that summarize the article. I would appreciate if you could provide this figure and the bullet points.

Thank you again for giving us the chance to consider your manuscript for The EMBO Journal, I look forward to hearing from you and receiving your final revised version of the manuscript.

Kind regards,

Daniel

>> Please limit the keywords for your study to maximally five.

>> Author Contributions: Please remove the author contributions information from the manuscript text. Note that CRediT has replaced the traditional author contributions section as of now because it offers a systematic machine-readable author contributions format that allows for more effective research assessment. and use the free text boxes beneath each contributing author's name to add specific details on the author's contribution.

More information is available in our guide to authors.
<https://www.embopress.org/page/journal/14602075/authorguide>

>> Adjust the title of the 'Competing Interests' section to 'Disclosure and Competing Interests Statement' and move after Acknowledgements. Please specify the authors who have filed the patent.

>> Provide a completed Author Checklist.

>> Please provide the main manuscript text as .docx file.

>> Figures in separate files: The main figures should be uploaded as individual, high res figure files. Up to 5 of the suppl. figures can be made EV figures: the figures should be uploaded as individual high res figure files and their legends should be in the manuscript text, after the main figure legends. The remaining figures should be compiled in a PDF labelled "Appendix"; their legends should be removed from the manuscript and displayed under each corresponding figure. The supplementary tables should also be added to the appendix and their legends should be removed from the manuscript text. Correct nomenclature is "Appendix Figure S1" etc. and "Appendix Table S1" etc. The appendix will need a table of contents including page numbers.

>> Provide source data for the study as to the separate request e-mail by my colleague Hannah Sonntag.

>> Funding: enter the funding information in the list of funders in our online system.

>> References: adjust the reference format to EMBO Journal format, 10 authors et al, and place References after the Discussion, before figure legends.

>> Correct order of manuscript sections: Abstract / Keywords / Introduction / Results / Discussion / Methods / Data Availability / Acknowledgements / Disclosure and competing interests statement // References / Figure legends / Tables and their legends / Expanded View Figure legends

>> Rename the "Experimental Procedures" section to "Methods".

>> The heading "Supplemental Information" should be changed to "Expanded View Figure Legends" and the figures should be changed to "Figure EV1" -EV5.

>> Add a Reagents and Tools table to the Methods section, as a separate file using the existing template in the Guide For Authors, listing key reagents, experimental models, software and relevant equipment.

>> Data availability section: remove referee tokens and make sure GEO and PRIDE datasets are made publicly accessible.

>> Recheck the bioRxiv citation Redo-Riveiro et al (2023) for journal publication and update the journal reference in case.

>> Add a separate 'Statistical Analysis' section to the Methods part, detailing the algorithms and statistical tests applied.

>> Consider additional changes and comments from our production team as indicated below:

*DATA CHECK: PASS

- DAS:

1. Please note that the specific URL for GSE173543 dataset is not provided in the data availability statement.
2. Please note that reviewer access code for GSE173543 dataset is provided in the manuscript.

- Figure legends:

1. Please note that the exact p values are not provided in the legends of figures 1E, F; 2C, D; 3A, H.
2. Please indicate the statistical test used for data analysis in the legend of figure 2A.
3. Please note that the box plots need to be defined in terms of minima, maxima, centre, bounds of box and whiskers, and percentile in the legends of figures 3A, H.
4. Please note that the box plots need to be defined in terms of minima, maxima and percentile in the legend of figure 2D.
5. Please note that information related to n is missing in the legends of figures 3A, H.
6. Please note that the error bars are not defined in the legends of figures 1B.
7. Please note that scale bar and its definition are missing for figures 1E, F; 4C.

>> note that the checks were only done for the main figures; please check the supplemental figures to ensure that similar issues there are also addressed

EMBOJ-2024-119954-T

Referee #1, additional comment:

'I have examined the responses to round 2 of review and the revised manuscript. I stand by my original decision on the revised manuscript, which was to recommend publication. The authors have addressed the key concerns in the first round of review, and added some more data in their rebuttal to provide further support concerning the role of FAO. I think that additional demands of Reviewer 4 and their comments in relation to Reviewer 2's points, whilst scientifically valid, are out of scope for this manuscript given its main conclusions and the significant additional data submitted in revision. As it stands, I think the manuscript is a strong contribution to ES cell biology and one that is worthy of the EMBO Journal.

The authors addressed the remaining editorial issues.

Dear Josh,

Thank you for submitting the revised version of your manuscript. I have now evaluated your amended manuscript and concluded that the remaining minor concerns have been sufficiently addressed.

I am thus pleased to inform you that your manuscript has been accepted for publication in the EMBO Journal.

Related, I would like to hereby ask your consent on keeping the referee figures included in this file.

On a different note, I would like to alert you that EMBO Press offers a format for a video-synopsis of work published with us, which essentially is a short, author-generated film explaining the core findings in hand drawings, and, as we believe, can be very useful to increase visibility of the work. Please see the following link for representative examples and their integration into the article web page:

<https://www.embopress.org/doi/full/10.15252/emj.2019103932>

Best regards,

Daniel

Daniel Klimmeck, PhD
Senior Editor
The EMBO Journal
EMBO
Postfach 1022-40
Meyerhofstrasse 1
D-69117 Heidelberg
contact@embojournal.org
